# The chitin synthase regulator CSR-3 promotes cellular integrity during cell-cell fusion in the filamentous ascomycete fungus *Neurospora crassa*

**Stephanie Herzog, Tanja N. Sedlacek, Kristian D. R. Roth[ID], Manuel Reuning, Ulrike Brandt[ID], André Fleißner[ID]***

Institut für Genetik, Technische Universität Braunschweig, Braunschweig, Germany

\* a.fleissner@tu-bs.de

## Abstract

Cell-cell fusion in plants and fungi requires localized cell wall dissolution at the contact site to allow direct plasma membrane contact and subsequent membrane merger. Since cell wall removal carries the risk of cell rupture, the process must be tightly regulated to permit localized fusion pore formation while preserving cellular integrity. While the molecular events guiding cell-cell signaling leading to contact between fusing fungal cells have begun to unfold, the post-contact mechanisms stabilizing the forming fusion pore remain largely unknown. Here, we identify the chitin synthase regulator CSR-3 as a molecular factor promoting stable pore formation during somatic fusion in the fungal cell fusion model *Neurospora crassa*. CSR-3 specifically accumulates at the contact zones of fusing cells and contributes to fusion fidelity by preventing membrane rupture and lysis, particularly under calcium-limited conditions. Loss of CSR-3 leads to elevated fusion-induced lysis, a phenotype rescued by osmotic stabilization, suggesting a cell wall defect. Beyond fusion, CSR-3 is involved in septum formation, septal pore plugging, conidiation, and the response to biotic and abiotic cell wall stress. These observations support a broader role for CSR-3 in chitin-mediated cell wall remodeling. Our data indicate that CSR-3 dynamics at fusion sites depend on the MAP kinase MAK-1, implicating cell wall integrity signaling in post-contact fusion events. Consistent with this finding, phospho-mimetic analysis suggests a regulatory role for CSR-3 phosphorylation. Co-localization and genetic analyses identify the chitin synthase CHS-2 as a likely downstream target of CSR-3, with both proteins functioning in the same pathway. Together, our findings reveal that CSR-3 coordinates cell wall remodeling during cell fusion and stress responses, uncovering a crucial regulatory layer that safeguards fungal cellular integrity during dynamic developmental processes. Our observations support a model in which cell wall biosynthesis plays a critical role in cell wall remodeling during fusion pore formation.

**Data availability statement:** All relevant data are within the manuscript and its Supporting information files.

**Funding:** This work has been partially supported by funding from the German Research Foundation (https://www.dfg.de/de) to AF (FL706/2-1 and FL706/3-1). The funders did not play any role in the study design, data collection and analysis, decision to publish, or preparation of the manuscript.

**Competing interests:** The authors have declared that no competing interests exist.

## Author summary

Cell-cell fusion is essential for the development of most eukaryotic organisms, yet its molecular mechanisms remain poorly understood. The ascomycete fungus *Neurospora crassa* serves as a valuable model for studying this fundamental process. In this study, we investigate how a stable fusion pore is formed between two merging fungal cells. We present the first experimental evidence that cell wall biosynthesis—specifically chitin synthesis—plays a critical role in establishing a stable fusion pore. Our findings suggest that cell wall remodeling during fusion involves a distinct set of enzymes compared to those active during general growth. We further show that the chitin synthase regulator *csr-3* is a target of the cell wall integrity MAP kinase pathway and likely regulates the chitin synthase CHS-2. This work advances our broader understanding of cell wall remodeling in fungal growth, development, and stress responses, and specifically highlights its role in cell-cell fusion. More broadly, our findings contribute to a better understanding of the cellular biology of filamentous fungi - an ecologically, industrially, and clinically important but still vastly underexplored group of organisms.

## Introduction

Cell-cell fusion is essential for a plethora of developmental and reproductive processes in eukaryotic organisms and occurs across all lineages of this taxonomic domain. Examples include gamete fusion during sexual propagation, somatic fusion in epidermal morphogenesis of nematodes, the merger of myoblasts with muscle founder cells during muscle development, or trophoblast fusion during placenta formation in mammals [1]. Despite its importance, the molecular mechanisms driving cell fusion are not fully understood, motivating numerous studies across diverse model organisms spanning the animal, plant, and fungal kingdoms. Fungal model systems include baker's yeast, fission yeast and the filamentous fungus *Neurospora crassa* [1,2]. In the two yeast species, fusion occurs only during mating between cells of opposite mating types. In contrast, the filamentous fungus *N. crassa* undergoes both sexual fusion during mating, and somatic fusion between genetically identical cells at two distinct stages of vegetative growth. During colony establishment, germinating spores attract each other, grow towards one another, and fuse, forming a supracellular structure that develops into the mycelial colony [3]. Later, within mature colonies, hyphal branches fuse to create cross-connections between leading hyphae, enhancing interconnectivity within the mycelial network [4]. Both fusion types are mediated by a unique signaling mechanism, where the two fusion cells likely take turns in signal sending and receiving [5,6]. The underlying signaling network mediating this kind of cellular dialog comprises the MAK-1 and MAK-2 MAP kinase pathways, homologous to the cell wall integrity and pheromone response pathways of baker's yeast, along with NADPH oxidase complexes and the conserved STRIPAK signaling complex [7].

While the mechanisms controlling cell-cell communication and directed growth during the early stages of fusion have started to unfold, the subsequent molecular processes governing post-contact fusion remain largely unexplored. Fusion pore formation following cell contact must be precisely regulated in both spatial and temporal dimensions, as malfunctions at this stage can lead to cell lysis and eventual death [8,9]. Cell fusion in fungi requires precise remodeling of the cell wall, a complex and unique structure that is essential for maintaining cellular integrity. The dissolution of the cell wall facilitates the fusion of plasma membranes, but this process must be tightly controlled to result in a stable fusion pore and to prevent cellular damage. The mechanisms contributing to fusion pore stabilization remain, however, mostly unknown. In *S. pombe*, cell wall degrading hydrolases are focused via actin and myosin V to the cell contact region, where the fusion pore will form in a spatially restricted, highly controlled manner [10].

Cell wall remodeling for fusion pore formation must involve all components of the fungal cell wall, which consists of a variety of interconnected polysaccharides and proteins. While cell wall compositions are taxon-specific, the fundamental building blocks — such as glucans, cell wall proteins, and chitin — are conserved across the fungal kingdom [11,12]. In many fungal species, including *Neurospora crassa*, chitin is predominantly found in the innermost layers of the cell wall, where it fulfills a crucial role in stability and shaping of the cells. Together with other cell wall components it provides wall integrity and protection while still enabling the plasticity required for cell growth and morphogenesis [13]. Chitin is synthesized by enzymatic transmembrane complexes known as chitin synthases (CHSs), whose activity is regulated by a network of associated activators and regulators [14]. The subcellular localization and activity of CHSs are controlled by these regulators, which mediate conformational changes, guide CHSs through the secretory pathway within chitosomes, and ensure its retention and activation at specific target locations [15,16]. *N. crassa* possesses seven chitin synthases (CHS-1 to CHS-7, each belonging to one of the seven defined classes) and three chitin synthase regulators - CSA-1 (NCU09322), CSA-2 (NCU02592), and CSR-3 (NCU02351) [17]. All three regulators are homologs of the chitin synthase regulator Skt5p (Chs4p) from *Saccharomyces cerevisiae*. As in other fungi, the different chitin synthases of *N. crassa* contribute to cell wall synthesis and remodeling and play distinct but sometimes redundant roles in various developmental processes, including hyphal tip extension, septation, spore formation, and stress response [17,18].

So far, no function of chitin synthases or their regulators has been reported for cell wall remodeling during fungal cell fusion. However, based on the assumption that fusion pore formation must be precisely controlled, we hypothesized in this study that not only cell wall deconstruction but also cell wall biosynthesis may be required for the controlled formation of a spatially restricted, stable fusion pore. To test this hypothesis, we analyzed the potential role of the chitin synthase regulators in somatic cell fusion in *N. crassa*. We show that the chitin synthase regulator CSR-3 contributes to fusion fidelity. In addition, we provide evidence that CSR-3 interacts with the chitin synthase CHS-2 (CHS class II) and that its regulatory activity during fusion pore formation is associated with the cell wall integrity MAP kinase signaling pathway. These findings offer new insights into the molecular mechanisms governing cell wall integrity during fusion and highlight the essential role of chitin synthesis in maintaining cell fusion fidelity in fungi.

## Results

### The chitin synthase regulators CSR-3 accumulates at cell-cell fusion sites

To test our hypothesis that cell wall synthesis contributes to fusion pore formation, we conducted a preliminary analysis of the subcellular localization of the three predicted chitin synthase regulators CSA-1, CSA-2, and CSR-3 in *N. crassa*. The three factors were independently fused with GFP and localized in germinating conidia. While all three proteins showed cytoplasmic localization in pre-contact cells, only CSR-3 accumulated at the cell-cell contact zone, where cell wall remodeling and plasma membrane fusion is taking place (S1 Fig). We therefore conducted a detailed functional characterization of this regulator.

The *csr-3* open reading frame (NCU02351) comprises 3342 bp including three predicted introns, whose positions we confirmed by cDNA sequencing. It encodes a protein of 935 amino acids (https://fungidb.org/) (S2A Fig), with 30% identity to the chitin synthase regulator Skt5p/Chs4p of *S. cerevisae.* A CDSEARCH/cdd analysis (BLAST) of CSR-3 revealed a TPR domain at aa position 185-466 with *Sel1*-like repeats, which might mediate protein-protein interaction, and a proposed rare atrophin-1 like domain (aa 469-888) with no equivalent within the other two SKt5p homologs CSA-1, CSA-2. At the C-terminus, a Caax box motif was predicted by PrepPS, suggesting potential membrane association of the protein by prenylation [19–21].

## CSR-3 is dispensable for general growth and development of *N. crassa* but promotes stable fusion pore formation

To test the role of CSR-3 in the growth and development of *N. crassa* with particular focus on cell-cell fusion, a Δ*csr-3* gene knockout mutant was created by using a gene replacement strategy (S2B, S2C Fig). For a control strain, the gene was reintroduced into this mutant. A general growth characterization revealed that the loss of *csr-3* did not affect macroscopic colony appearance, sporulation or the formation of aerial hyphae (S3A, S3D, S3E Fig). Linear hyphal growth was, however, slightly decreased and cultures in glass tubes produced a characteristic brown pigment at the hyphal/glass interphase. Both phenotypes were absent in the complemented isolate (S3B, S3C Fig).

Spore germling interactions during colony establishment were unaffected in the *csr-3* gene knockout strains at all precontact stages, including spore germination, cell-cell recognition and directed growth (S4 Fig). To determine cell-cell fusion, spores expressing either cytoplasmic GFP or mCherry were mixed, and fusion pairs consisting of a red and green fluorescent cell were analyzed. In case of successful fusion, the two different fluorescence signals mix and appear in both cells, while fusion failure is indicated by the continuous separation of the two signals. Mutant analysis revealed that CSR-3 is dispensable for cell-cell merger, despite its accumulation at the fusion side (Fig 1A).

Based on its predicted molecular function as a chitin synthase regulator, we hypothesized that CSR-3 might rather be involved in stabilization of the forming fusion pore than in cell wall unsealing during pore formation. We reasoned that deficiencies in pore stabilization might result in membrane rupture and cell lysis during the fusion process. To test this hypothesis, we assessed lysis of fusion pairs. Earlier studies indicated that membrane rupture and subsequent cell lysis are potential outcomes of aberrant cell fusion, detectable by a strongly vacuolized appearance of lysed cell pairs [8,9,22]. Depletion of calcium from the growth medium usually increases the lysis rate, probably due to an inhibition of calcium-dependent membrane repair mechanisms [9,23]. The quantification of cell lysis revealed a significant increase in the Δ*csr-3* mutant compared to the wild type or the complemented strain. This difference was even more pronounced on medium with reduced amounts of calcium, where about 16% of the mutant cell pairs lysed (Fig 1B, 1C).

In order to distinguish whether this lysis phenotype is caused by a cell wall or a plasma membrane defect, the medium was osmotically stabilized. This stabilization would rescue cell wall deficiencies but not lysis primarily caused by membrane defects. Consistent with a predicted role of CSR-3 in cell wall remodeling, the lysis phenotype of the mutant could be suppressed by the addition of 1M sorbitol to the growth medium (Fig 1D).

Taken together these data indicate that CSR-3 contributes to fidelity of the cell fusion process during germling fusion, probably by stabilizing the forming fusion pore.

## CSR-3 localizes around the expanding fusion pore in germling and hyphal fusion

To conduct a more detailed analysis of the subcellular CSR-3 dynamics and to test the functionality of the employed CSR-3-GFP construct, the expression construct used in the initial localization study was introduced into the Δ*csr-3* mutant. Despite a distinct GFP signal, the lysis phenotype of the mutant was not complemented by this construct, suggesting that C-terminal tagging of CSR-3 interferes with the proteins function. We therefore created alternative strains, expressing

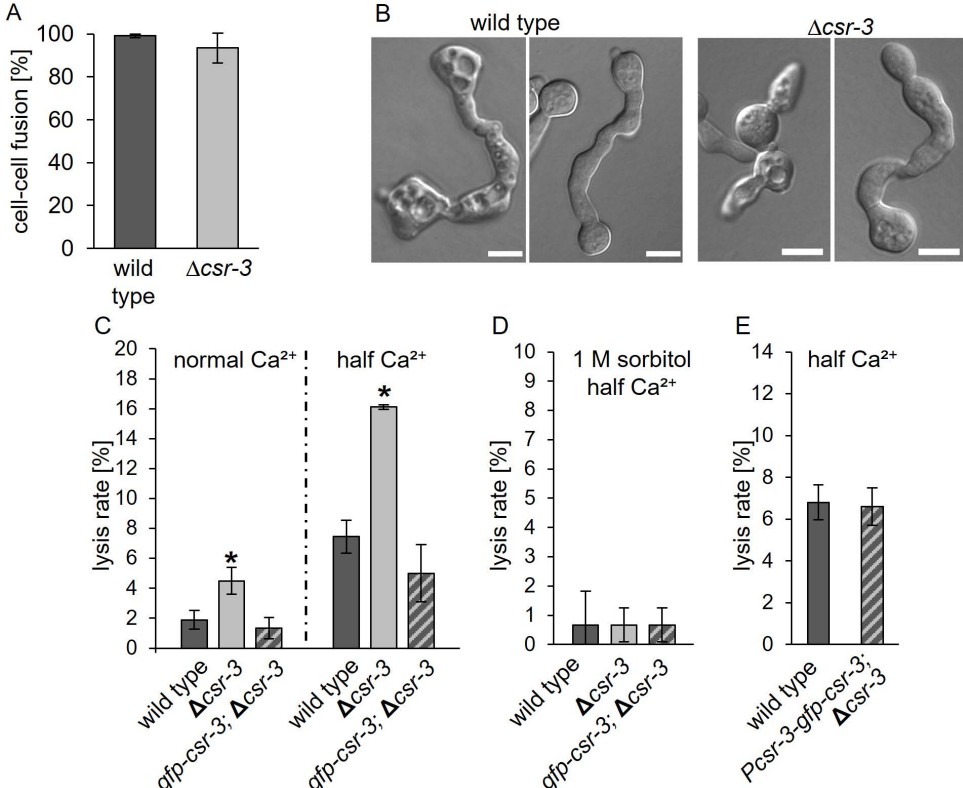

**Fig 1. CSR-3 promotes stable fusion pore formation. (A)** Quantification of successfully fused germling pairs of the wild-type strain (FGSC 2489) and the Δ*csr-3* mutant (GN5-20) after 4 h of incubation (for details see material and methods). **(B)** Lysed germling pairs of the wild-type strain (FGSC 2489) and the Δ*csr-3* mutant (GN5-20) are characterized by a vacuolized appearance (left) in contrast successfully fused pairs (right). Scale bars: 5 µm **(C-E)** Quantification of lysis rate during germling fusion of wild type (FGSC 2489), the Δ*csr-3* mutant (GN5-20) and complemented strains (SH_59: P*ccg-1-gfp-csr-3*, Δ*csr-3* and SH_283: P*csr-3-gfp-csr-3*, Δ*csr-3*) on MM and MM with half Ca²⁺ (A,E) and under membrane stabilizing conditions **(D)**. Error bars represent the standard deviation from at least three independent experiments (A) n = 180 to 220 pairs, C,D,E n = 3x100 each). Asterisks indicate statistically significant differences (p ≤ 0.05) compared to reference strain (FGSC 2489).

N-terminally GFP-tagged CSR-3 under control of the native (P*csr-3*) or the *ccg-1* promoter, which is routinely used for protein localization studies in *N. crassa* [24]. Both variants fully complemented the lysis phenotype and the slight growth reduction of Δ*csr-3*, indicating their full functionality (Figs 1E and S3). In fusing spore germlings of both strains, GFP-CSR-3 accumulated at the cell contact site. However, expression from the native promoter yielded only very weak signals unsuited for subsequent comprehensive localization studies (S5 Fig). All following experiments were therefore conducted with the strain, carrying the P*ccg-1* expression construct. Time-lapse microscopy revealed that CSR-3 is evenly distributed throughout the cytoplasm in non-interacting cells and during the tropic interaction of fusion partners, without specific association to organelles. Importantly, the protein does not accumulate at the expanding germ tube tips. However, as soon as the germ tubes establish physical contact, GFP-CSR-3 accumulates at the contact sites, where it remains until cell fusion is completed. In time lapse, the signal appeared as an expanding ring, suggesting that CSR-3 localizes around the forming fusion pore (Fig 2A–2C and S1 Movie).

In *N. crassa*, vegetative fusion is not limited to germlings, but also occurs between hyphal branches in the inner parts of the mature mycelial colony [4]. Live cell imaging of 34 hours old mature colonies, revealed that GFP-CSR-3 also accumulates at the hyphal fusion contact sites in an expanding ring-like structure (Fig 2D, 2E).

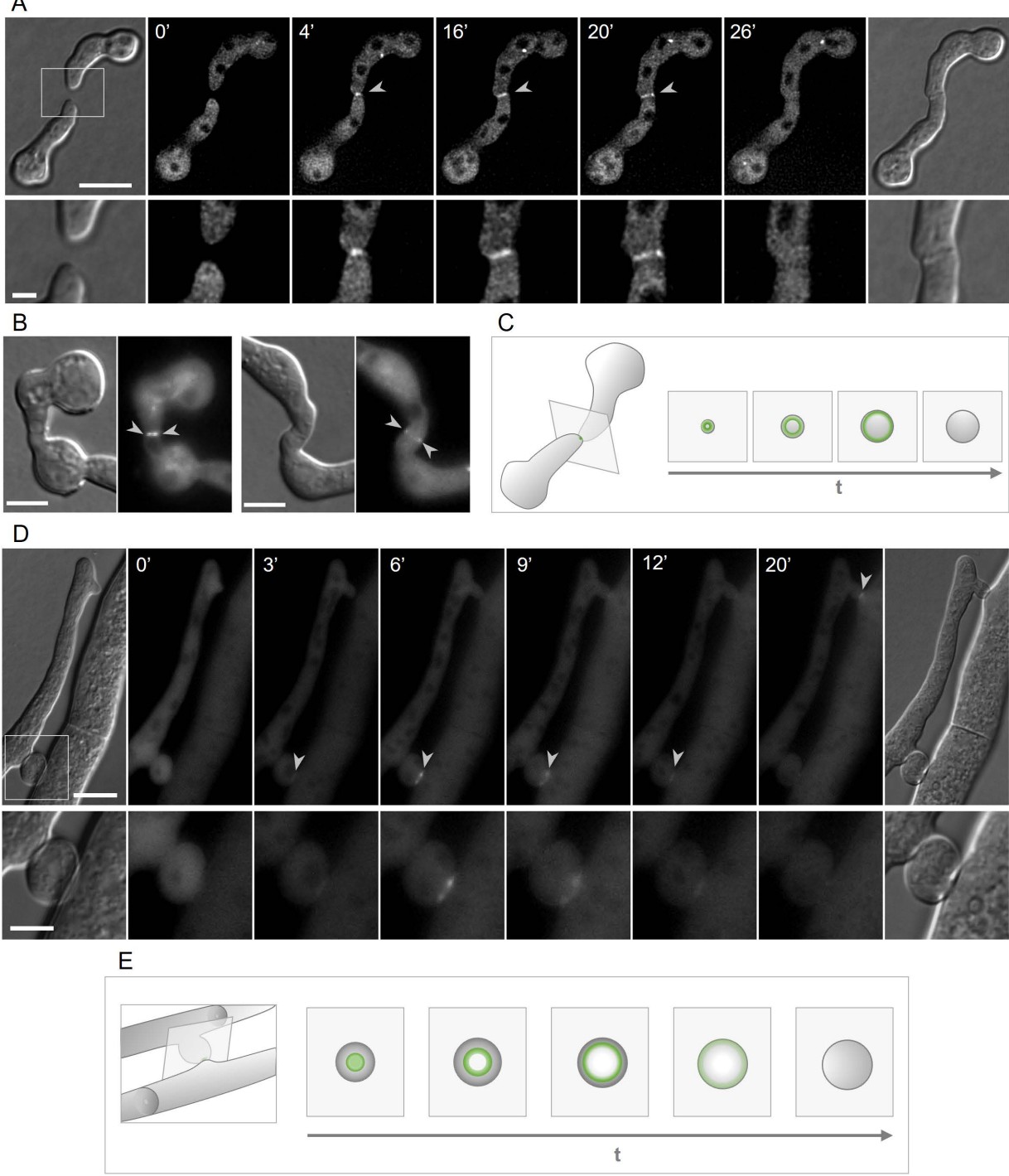

**Fig 2. CSR-3 localizes in an expanding ring-like structure during vegetative fusion. (A)** Subcellular localization of GFP-CSR-3 in strain SH_125 (*Pccg-1-gfp-csr-3*, Δ*csr-3*) during germling fusion (for corresponding movie see S1 Movie). Scale bar: 10 μm (overview) and 1 μm (inset). **(B)** Lateral sections of fusing germling pairs of strain SH_125 within an early (left) and late (right) fusion state. Scale bars: 5 μm. **(C)** Schematic illustration of the localization pattern of CSR-3 during germling fusion. **(D)** Time course of GFP-CSR-3 in strain SH_59 (*Pccg-1-gfp-csr-3*, Δ*csr-3*) during hyphal fusion. Scale bar: 10 μm (overview) and 2 μm (inset). **(E)** Schematic illustration of the localization pattern of CSR-3 during hyphal fusion.

In summary, CSR-3 shows a specific accumulation around developing cell fusion pores in germling and hyphal fusion consistent with a proposed function in fusion pore stabilization by chitin synthesis.

## CSR-3 contributes to septa formation, conidiation and septal pore plugging

To determine whether CSR-3 function is specific to cell-cell fusion or also involved in other developmental processes, we examined its subcellular localization during other growth stages that require chitin synthesis. These stages included hyphal tip extension, septa formation, septal pore plugging after injury, and macro- and microconidia formation. No CSR-3-GFP signal was detected at the tips of actively growing hyphae, suggesting that CSR-3 does not contribute to hyphal extension. In contrast, an intense signal was detected at all forming septa observed during this analysis (n = 45). Time lapse revealed a contracting ring-like structure at the leading edge of the forming septum (Fig 3A–3D, 3F). Comparable observations were made during septum formation in spore germlings, suggesting a comparable function of CSR-3 in both processes (Fig 3E, 3F). Mutant analysis revealed, however, that CSR-3 is dispensable for septa formation in hyphae or germlings (S6 Fig).

CSR-3 also accumulated at the majority of septal plugs formed in response to mechanical hyphal injury induced by razor blade cutting. (94,2 ± 1,5%, n > 60) (S13A, S13B Fig). An earlier study reported *de novo* chitin deposition at these pore sealing structures, which prevent extensive cytoplasmic loss [25].

*N. crassa* employs two different modes of cell division for the formation of two types of conidia [26]. Multinucleate macroconidia are formed by conidiophores via a fission-like cell division, which involves the formation of a double septum [27]. In contrast, uninucleate microconidia originate from hyphae through a budding-like process [28]. GFP-CSR-3 contributes to both processes and localizes to the double septum of macroconidia and the bud neck during microconidia formation (S7 Fig).

In summary, CSR-3 contributes to various cellular processes, requiring chitin synthesis. This observation strongly supports the hypothesis that CSR-3 is controlling chitin synthase activity. However, it is not detected at growing germ tube or hyphal tips, suggesting that cell wall remodeling during cell tip growth and fusion relies on different sets of enzymatic activities.

## CSR-3 contributes to the cellular response against cell wall targeting compounds

To test if CSR-3 might also contribute to the cellular response towards cell wall injury, we analyzed the subcellular dynamics of CSR-3 in the presence of Lysing Enzyme, a commercially available mix of cell wall degrading enzymes produced by the mycoparasitic fungus *Trichoderma spec.*, which comprises cellulase, glucanase, protease and chitinase activity. Conidia expressing GFP-CSR-3 (strain SH_125) were incubated in liquid medium for three hours to induce germination, before lysing enzyme was added to the cultures. The spore germlings were analyzed by light and fluorescence microscopy directly before and after treatment with the enzyme mix. In the absence of lysing enzyme GFP-CSR-3 located within the cytosol of the healthy fungal cells, as described before. Within two minutes after the addition of lysing enzyme, cell wall integrity became strongly affected, indicated by the observable loss of cytosol at germ tube tips (S8 Fig). GFP-CSR-3 strongly accumulated at the cell periphery of about one third of the cells (38% ± 4%) and specifically at the sites of visible injury, which often were located at cell tips of the germling (20% ± 4% of the cells) (S8E Fig). The applied concentration of lysing enzyme caused cell death in approximately every second germling after five minutes of incubation independent of the observed recruitment pattern. Comparable subcellular dynamics were never observed in control assays, where only the solvent for lysing enzyme was added.

The observed loss of cytoplasm during treatment with lysing enzyme, indicates that not only the cell wall but also the plasma membrane is damaged. To test if membrane damage alone might also induce membrane recruitment of CSR-3, we decided to test as a control the effect of the membrane disturbing compound tomatine on the GFP-CSR-3 expressing

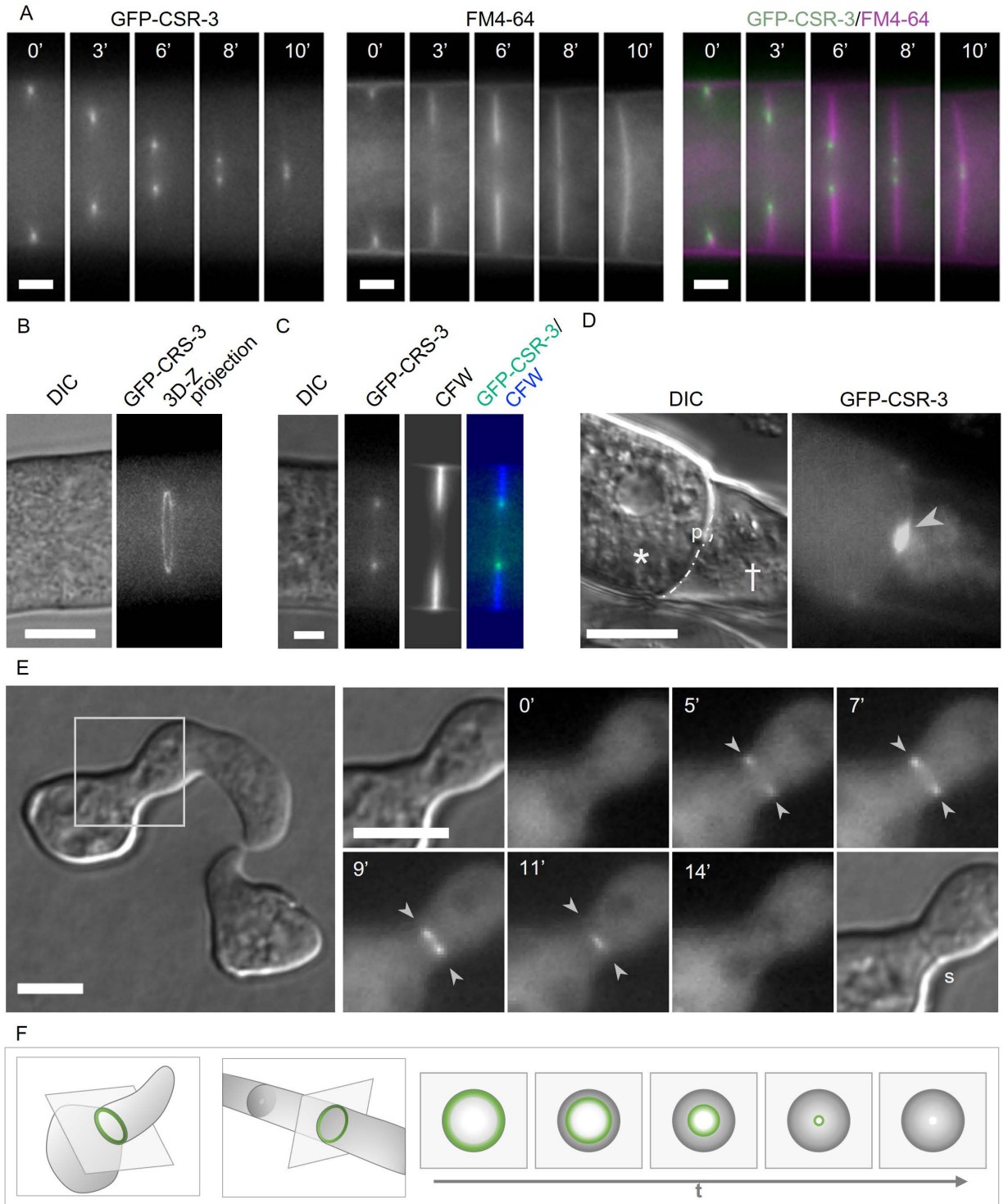

**Fig 3. CSR-3 localizes at developing septa in germlings and hyphae. (A)** Recruitment of GFP-CSR-3 to the inner ring of a forming septum in a hypha over time. Membrane was stained with FM4-64. **(B)** 3D-Z projection (25 individual images) of GFP-CSR-3 during septum formation. **(C)** Localization of GFP-CSR-3 during septation. Cell wall was stained CFW. **(D)** GFP-CSR-3 recruitment to the plug separating healthy (asterisk) and injured (cross) hypha parts **(E)** Localization of GFP-CSR-3 (arrow heads) during the septa formation (s) in a germling. **(F)** Illustration of the localization pattern of CSR-3 during septa formation. Used strains: SH_59 (*Pccg-1-gfp-csr-3*, Δ*csr-3*) (A-E) and SH_125 (*Pccg-1-gfp-csr-3*, Δcsr-3) (E). Scale bars: 2 µm (A), 3µm (C), 5 µm (B,E), 10 µm (D).

strain. The phytoanticipine tomatine forms non reversible pores in the membrane by binding to ergosterol. While no cytosolic loss was observed under tomatine treatment, around 75% of the germlings displayed a highly vacuolized appearance after five minutes of incubation indicating that these cells died from the treatment with the phytoanticipine (S8 Fig). More than 50% of all germlings also displayed a clear accumulation of GFP-CSR-3, however, exclusively at the germ tube tips (53% ± 10% of all cells).

Since the quantifications only represent snapshots of a highly dynamic process, we monitored GFP-CSR-3 recruitment in germlings in presence of lysing enzyme and tomatine over time. Therefore the substances were added to three hours old liquid cultures and cells were observed for at least 30 minutes by bright field and fluorescence microscopy. Like on solid agar plates, lysing enzyme caused rupture of the cells' plasma membranes indicated by the loss of cytosol (Fig 4 and S2 Movie). Again GFP-CSR-3 accumulated at the injured areas, and, remarkably, the protein was released from these spots after some minutes. In presence of tomatine, when cells began to vacuolize GFP-CSR-3 was recruited to the germ tube tips and after a while the signal vanished (S9 Fig). While the germlings often appeared to recover from the stress caused by lysing enzyme, as indicated by a decreasing number of vacuoles, no such recovery was observed with tomatine.

Since tomatine causes a severe effect on the fungus and CSR-3 displayed a subcellular reaction in presence of the phytoanticipine, we checked a putative role of the regulator in resistance on the macroscopic level. However, none of the tested tomatine concentration affected the growth of the Δcsr-3 mutant compared to the reference wild-type strain (FGSC 2489) (S10B Fig).

Based on the observation that CSR-3 accumulates at wound sites caused by lysing enzyme, we hypothesized that the protein might play a role in *N. crassa's* defense response to the lysing enzyme producing mycoparasite *Trichoderma artoviride*. To test this hypothesis, we mixed and co-cultivated conidia of *T. atroviride* with either spores of the *N. crassa* wild type (FGSC 2489), the Δcsr-3 mutant (GN5-20) or the complemented mutant (SH_ 283: P*csr-3-gfp-csr-3*, Δcsr-3) strain on VMM agar plates (S11A, S11B Fig). To minimize interference from calcium-dependent repair mechanisms, co-cultivations were also performed on medium containing half the standard calcium concentration (S11C, S11D Fig). After four days, colonies of both strains had formed on every plate and established physical contact. A degradation zone consistently developed in the *N. crassa* mycelium at the interface with the *T. atroviride* colony, characterized by reduced biomass and absence of conidiation. As a consequence, a nearly mycelia free ring surrounding the *T. atroviride* colony was formed. Under both tested media conditions, this lysis area was more pronounced in the Δcsr-3 mutant compared to the reference strains. The strongest difference could be observed after four days of incubation. Taken together, these observations suggest that CSR-3 does indeed contribute to the defense of *N. crassa* against the mycoparasite.

## The prenylation motif at the c-terminus is required for proper function of CSR-3 during fusion pore formation

As described above, fusion of the C-terminus of CSR-3 with GFP interfered with the function of the protein. Sequence analysis indicated a potential Caax box at the C-terminus that might result in post transcriptional prenylation (S2A Fig). Since this modification can mediate the interaction with lipid bilayers, we hypothesized that the C-terminus of CSR-3 might contribute to the proper spatial dynamics of this protein. To pinpoint this defect to the Caax box at the C-terminus, we created a CSR-3 version, in which the critical cysteine of the motif at aa position 932 is replaced by a serine residue, thereby preventing prenylation. The respective expression construct *gfp-csr-3^SaaX^* was expressed under the control of the native promotor (SH_290) and the constitutive P*ccg-1* promotor (SH_316) in the *csr-3* knock out background for lysis assays and localization studies. The localization of CSR-3^SaaX^ during septation and cell-cell fusion was unaffected on normal MM (Fig 5A–5C). In contrast, accumulation of CSR-3 at the fusion point was significantly reduced and the number of lysed germling pairs was significantly increased compared to the reference strain when external calcium was reduced by half (Fig 5C, 5D). Together, these data indicate that the Caax motif critically contributes to the recruitment and proper functioning of CSR-3 during fusion pore formation.

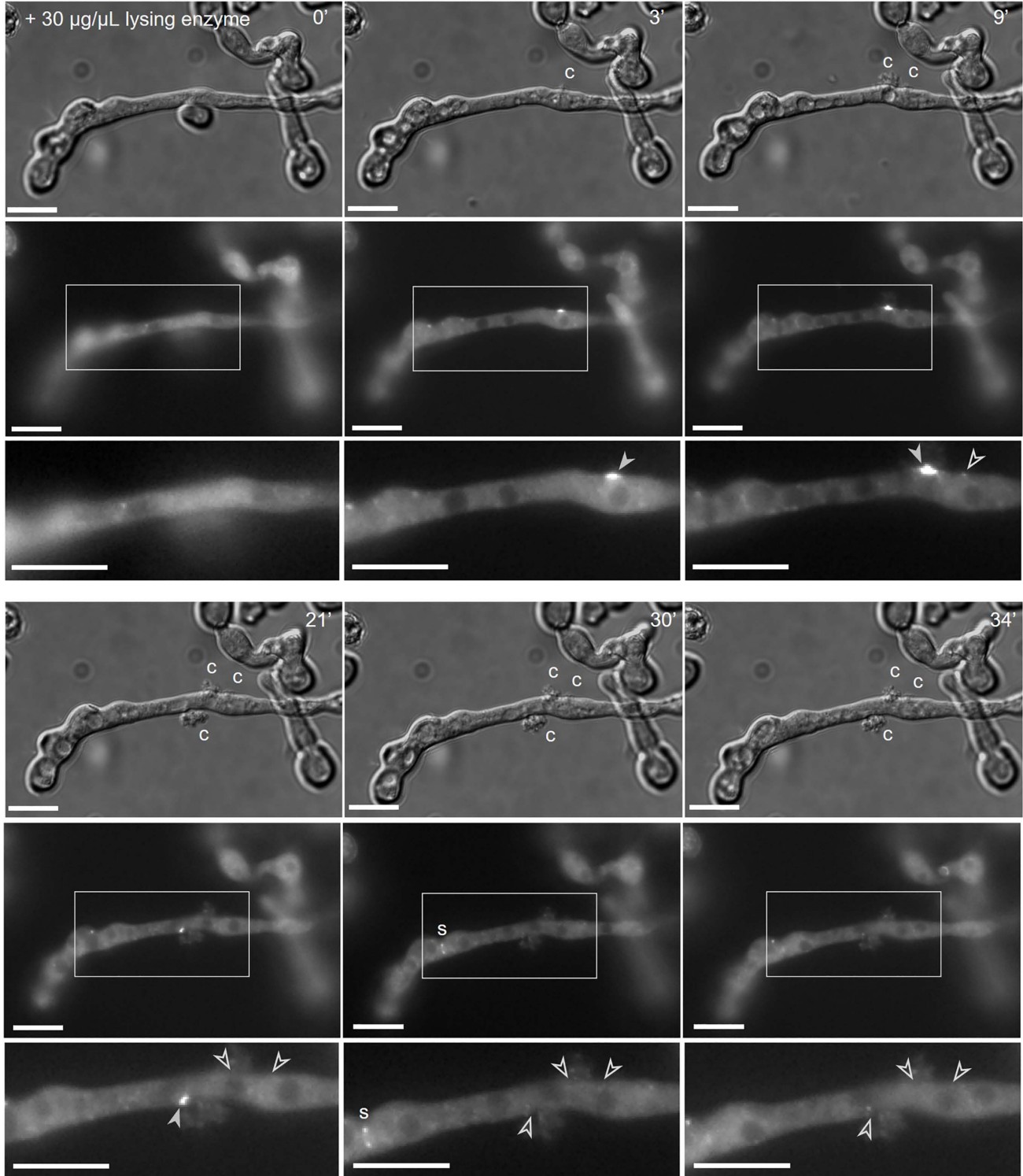

**Fig 4. CSR-3 localization is highly dynamic during treatment with lysing enzyme.** Time lapse of the subcellular localization of GFP-CSR-3 in a liquid culture (arrow heads; SH_125: *Pccg-1-gfp-csr-3*, *Δcsr-3*) after treatment with lysing enzyme (for corresponding movie see S2 Movie). Cytosolic leakages are marked with a c in the bright field images. Filled arrowheads indicate GFP-CSR-3 recruitment, empty ones mark locations where CSR-3 has already been released. A forming septum (s) is observed at time point 30'. Scale bar: 5 μm.

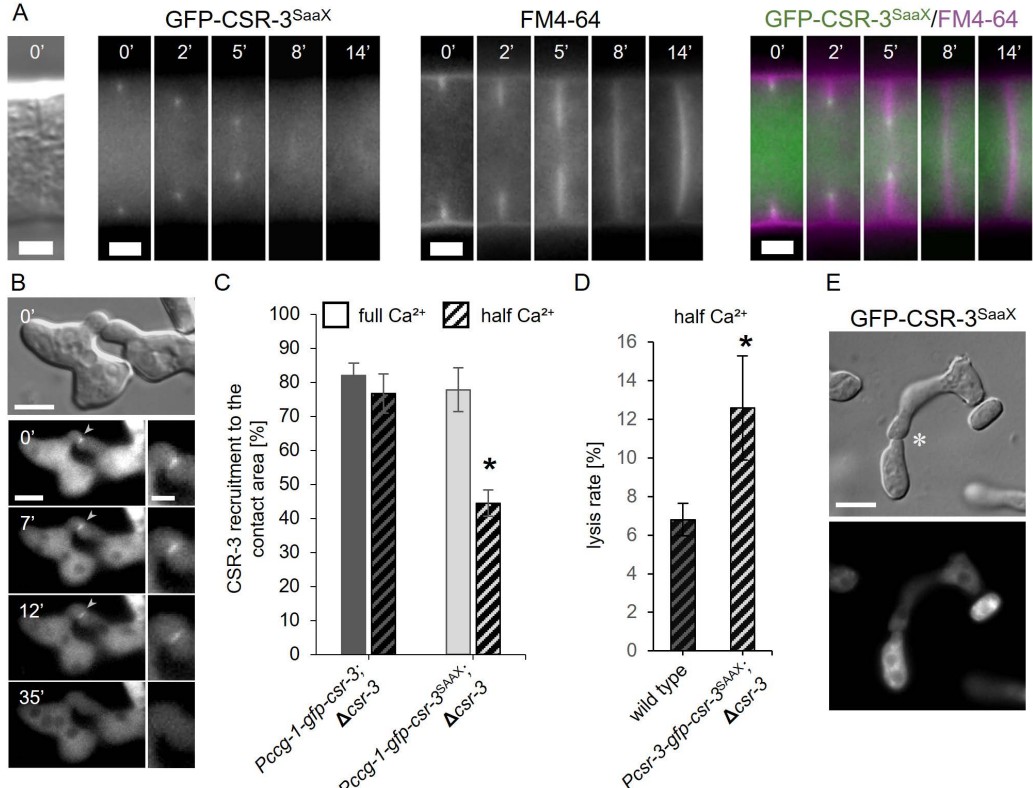

**Fig 5. The c-terminus of CSR-3 is important for proper function during fusion pore formation. (A)** Localization of GFP-CSR-3$^{SAAX}$ during septation in a hypha of strain SH_316 (*Pccg-1-gfp-csr-3*$^{SAAX}$, Δ*csr-3*). Membrane was stained with FM4-64. **(B)** Recruitment of GFP-CSR-3$^{SAAX}$ during germling fusion of strain SH_316 (*Pccg-1-gfp-csr-3*$^{SAAX}$, Δ*csr-3*). **(C)** Quantification of GFP-CSR-3 (SH_125) and GFP-CSR-3$^{SAAX}$ (SH_316) recruitment to the contact sites of fusing germlings on media with different external calcium concentrations. **(D)** Quantification of the lysis rate during germling fusion of the wild type strain (FGSC 2489) and the Δ*csr-3* mutant expressing GFP-CSR-3$^{SAAX}$ (SH_290: *Pcsr-3-gfp-csr-3*$^{SAAX}$, Δ*csr-3*) on MM with half Ca$^{2+}$. **(E)** Germlings expressing GFP-CSR-3$^{SAAX}$ (SH_316) show no recruitment of the protein to the contact sites (asterisk). Error bars represent standard deviation from at least three independent experiments (E: n = 15 to 20 pairs each; F: n = 3x100). Asterisks indicate statistically significant differences (p ≤ 0.05) compared to the reference strain. Scale bars: 3 μm (A); 5 μm (B,E overviews) and 2 μm (B inset).

## The three tested phosphorylation sites are dispensable for the recruitment of CSR-3 but might influence its function

An earlier phosphoproteome study identified three phosphorylation sites of CSR-3 [29], prompting us to test the role of these sites for the subcellular dynamics and function of the protein. We created two variations of CSR-3 with mutated phosphorylation sites that either prevent the residues to be phosphorylated (GFP-CSR-3$^{VAA}$) or imitate permanent phosphorylation by phosphomimetics (GFP-CSR-3$^{EDD}$) (for details see material and method section). Both constructs were expressed under the control of the native promotor (SH_310 and SH_314) and the P*ccg-1* promotor (SH_171 and SH_176) in Δ*csr-3* in order to quantify fusion pair lysis and to test the subcellular localization. The localization studies in hyphae and germlings of the strains SH_310 (*Pccg-1-gfp-csr-3*$^{VAA}$, Δ*csr-3*) and SH_314 (*Pccg-1-gfp-csr-3*$^{EDD}$, Δ*csr-3*) exposed no altered recruitment patterns during plugging and septation in hyphae, and fusion pore formation in germlings compared to the non-mutated version of CSR-3 (SH_125: *Pccg-1-gfp-csr-3*, Δ*csr-3*) (Fig 6A–6D). Similarly, a quantitative analysis of GFP-CSR-3$^{VAA}$ recruitment to the contact sites of fusing germlings found no significant differences to the reference strain SH_125 (Fig 6E). Despite this normal localization, germlings expressing GFP-CSR-3$^{VAA}$ (SH_171) lysed more

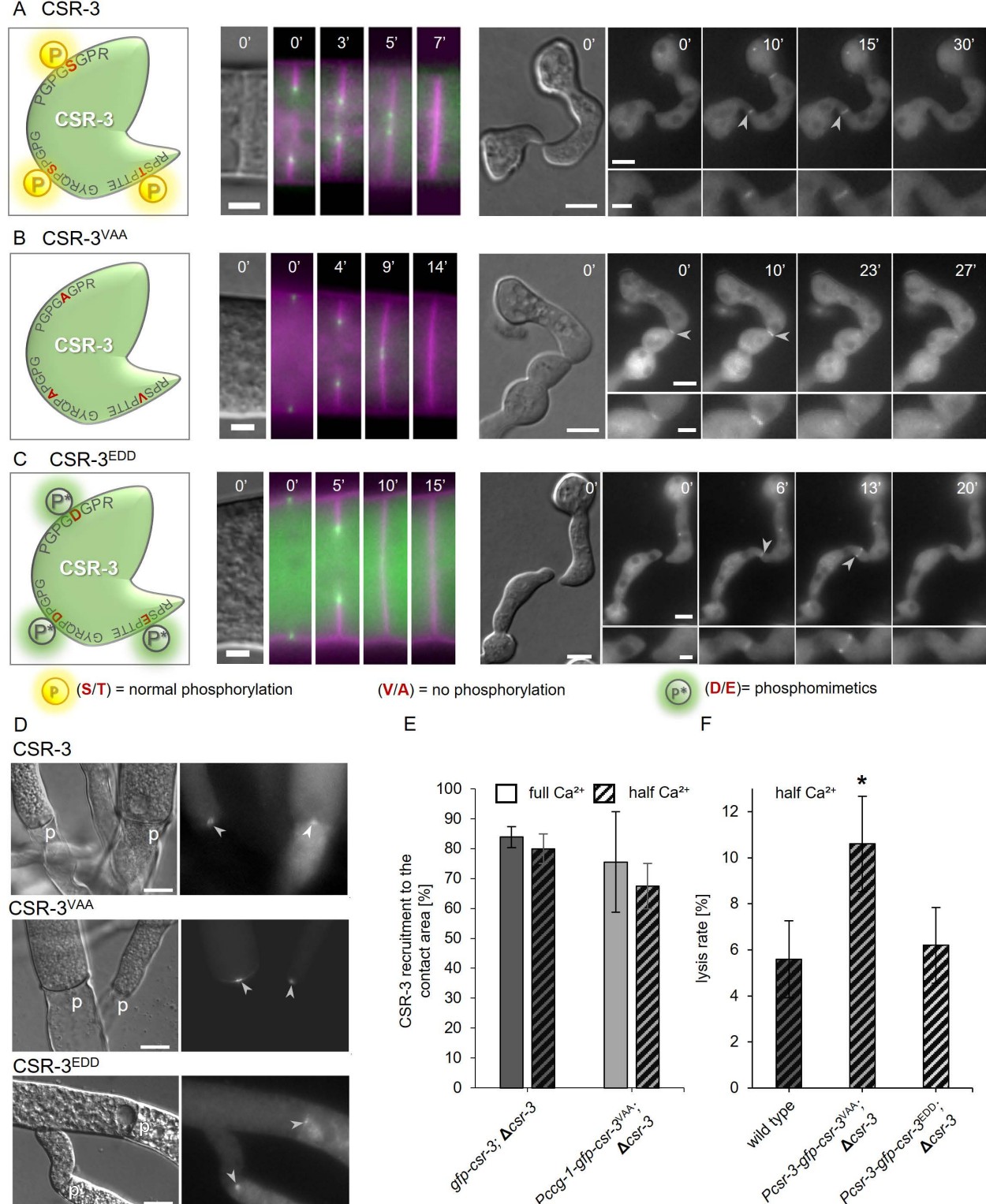

**Fig 6. Phosphorylation of CSR-3 influences its proper function during fusion pore formation. (A–C)**: Localization of different GFP-CSR-3 variants (left) with mutated phosphorylation sites. All three constructs are still recruited to the inner ring of forming septa (middle) and to the contact sites of fusing germlings (right) (used strains: SH_59: *Pccg-1-gfp-csr-3*, Δ*csr-3*; SH_171: *Pccg-1-gfp-csr-3VAA*, Δ*csr-3*; SH_176: *Pccg-1-gfp-csr-3EDD*, Δ*csr-3*).

Membranes were stained with FM4-64. **(D)** Recruitment of GFP-CSR-3 variations with mutated phosphorylation sites to the plugs (p) of injured hyphae (strains top to bottom: SH_59: *Pccg-1-gfp-csr-3*, Δ*csr-3*; SH_171: *Pccg-1-gfp-csr-3*$^{VAA}$, Δ*csr-3*; SH_176: *gfp-csr-3*$^{EDD}$, Δ*csr-3*). **(E)** Quantification of CSR-3$^{VAA}$ and of GFP-CSR-3 localization at the contact sites of fusing germlings in the strains SH_171 (*Pccg-1-gfp-csr-3*$^{VAA}$, Δ*csr-3*) and SH_125 (*Pccg-1-gfp-csr-3*, Δ*csr-3*) on MM and MM with half Ca$^{2+}$. **(F)** Quantification of the lysis rate during germling fusion of the wild type strain (FGSC 2489) and Δ*csr-3* mutant strains expressing either GFP-CSR-3$^{VAA}$ (SH_310: *Pcsr-3-gfp-csr-3*$^{VAA}$, Δ*csr-3*) or GFP-CSR-3$^{EDD}$ (SH_314: *Pcsr-3-gfp-csr-3*$^{EDD}$, Δ*csr-3*) on MM with half Ca$^{2+}$. Error bars represent the standard deviation from at least three independent experiments (E: n = 15 pairs each; F: n = 100 each). Asterisks indicate statistically significant differences (p ≤ 0.05) compared to the reference strains. Scale bars: 3 μm (A-C middle); 5 μm (A-C overview right); 2 μm (A-C inset right) 10 μm (D).

frequently during fusion (Fig 6F) on MM with with reduced calcium, whereas GFP-CSR-3$^{EDD}$ complemented the lysis phenotype of the Δ*csr-3* mutant. Taken together, these data indicate that while phosphorylation of the aa residues at position 166, 683 and 692 is dispensable for the subcellular dynamics of CSR-3, it is crucial for proper functioning of CSR-3 during fusion pore formation.

## The maintenance of CSR-3 at the contact sites of interacting germlings depends on MAK-1 activity

Although the molecular mechanisms mediating the late steps of the germling fusion process remain mostly elusive, some factors that accumulate at the cell-cell contact sites have been identified in earlier studies. Similar to CSR-3, the mitogen-activated protein kinase (MAPK) MAK-1 is recruited to the contact sites of interacting *N. crassa* germlings after cell-cell contact [30]. MAK-1 is part of a three-tiered cell wall integrity (CWI) MAP kinase module, comprising the MAPK MAK-1, the MAPK kinase (MAPKK) MEK-1 and the MAPKK kinase MIK-1. In *Sordaria macrospora*, Pro40 was identified as a scaffold protein of this MAP kinase module [31]. Its homolog in *N. crassa,* the SO protein, is also essential for germling fusion and also accumulates to the contact sites of fusing germlings [5]. To test a potential functional relationship between the cell wall integrity signaling pathway and CSR-3, we set out to colocalize these factors. In a first step, we confirmed the accumulation of the kinase MAK-1 at the contact sites and showed the same localization pattern for both its upstream kinases (Fig 7). When co-localized with CSR-3, both MAK-1 and SO localized together with the chitin synthase regulator throughout the entire duration of fusion pore formation (Fig 8). Based on these observations and above described finding that phosphorylation is needed for the proper CSR-3 functioning during fusion pore formation, we hypothesized that MAK-1 kinase activity is required for proper CSR-3 dynamics. To test this hypothesis, we employed a chemical genetics approach, in which the kinase activity of an ATP-analog-sensitive MAK-1 variant can be rapidly and specifically inhibited by the addition of the ATP analog 1-NM-PP-1 [6,30]. To test the effect of MAK-1 inhibition on the dynamics of the chitin synthase regulator, we expressed GFP-CSR-3 in the MAK-1 analog-sensitive mutant. In a parallel approach, we analyzed the dynamics of a GFP-tagged variant of the analog-sensitive MAK-1 MAP kinase. In control assays, we confirmed that the inhibitor has no effect on the protein dynamics in a MAK-1 wild-type background and that the solvent of the inhibitor did not cause any effects on the inhibitable strains (Fig 9).

Since MAK-1 plays various roles at the different stages of the fusion process, we restricted our analysis to cell pairs that did not display a signal at cell tips before inhibition and whose germ tubes were less than 10 μm apart but still had not established physical contact (S12 Fig). After addition of the inhibitor, these cell pairs usually still established physical contact, however, 90% failed to fuse. In about 41% and 31% of the germling pairs, GFP-CSR-3 and MAK-1$^{E104G}$-GFP, respectively, still localized to the contact sites of the two germ tubes for a short period of time, although none of the examined pairs completed fusion. We found no statistical difference between the localization patterns, suggesting a putative dependence of CSR-3 recruitment on MAK-1 kinase activity.

Taken together, our data provide first evidence that the dynamics of CSR-3 are at least in parts controlled by the cell wall integrity MAP kinase pathway.

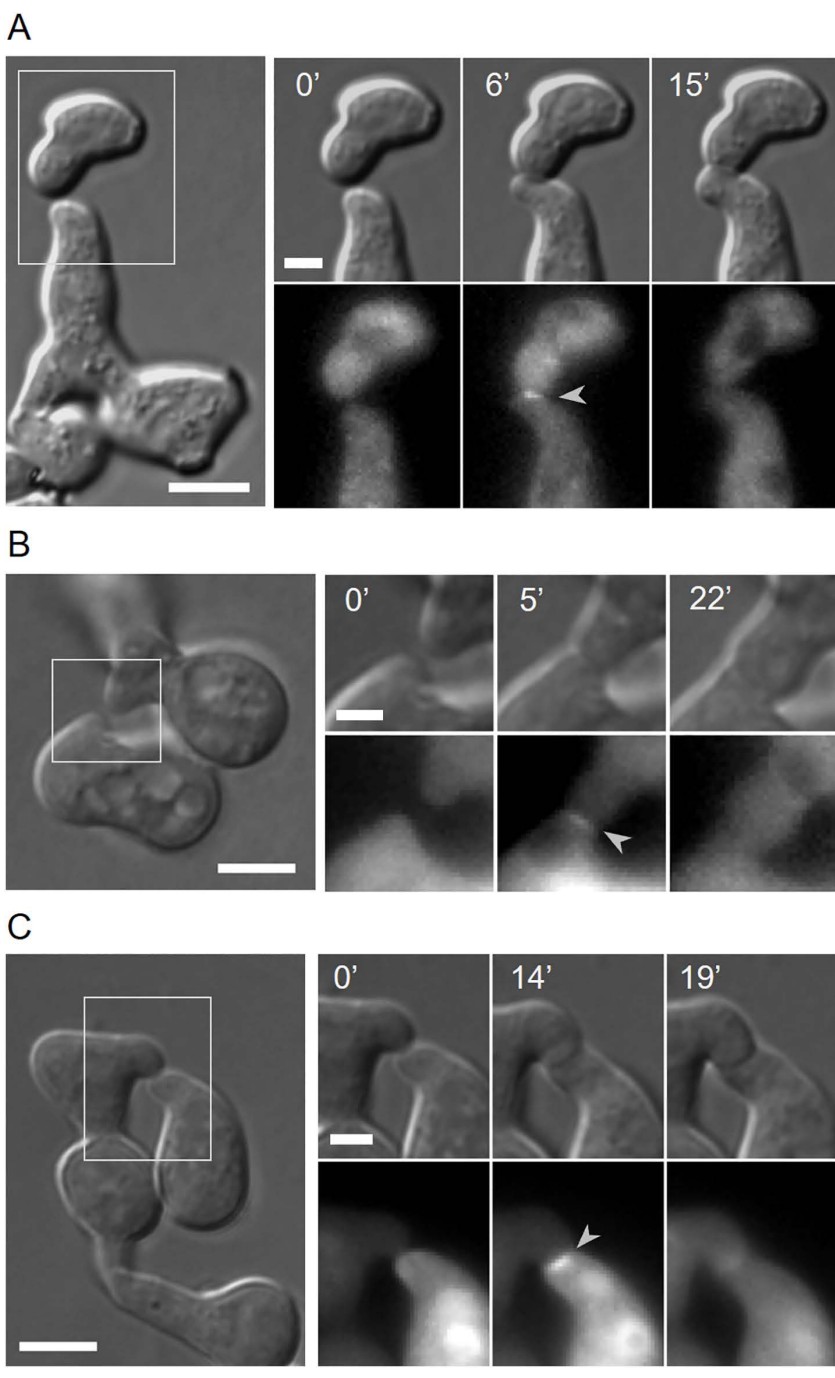

**Fig 7. All three MAP kinases of the MAK-1 kinase cascade localize at the contact site of fusing germlings. (A)** Localization of GFP-MIK-1 (arrow head) during germling fusion in the strain SH_261 (*Pccg-1-gfp-8xgly-mik-1*, Δ*mik-1*). **(B)** Subcellular localization of MEK-1-GFP (arrow head) in fusing germlings of the strain MW_581 (*Pccg-1-mek-1-gfp*, Δ*mek-1*). **(C)** Localization of GFP-MAK-1 (arrow head) during cell-cell fusion in germlings of the strain GN4-32 (*Pccg-1-gfp-mak-1*, Δ*mak-1*). Scale bars: 5 μm (overview) and 2 μm (inset).

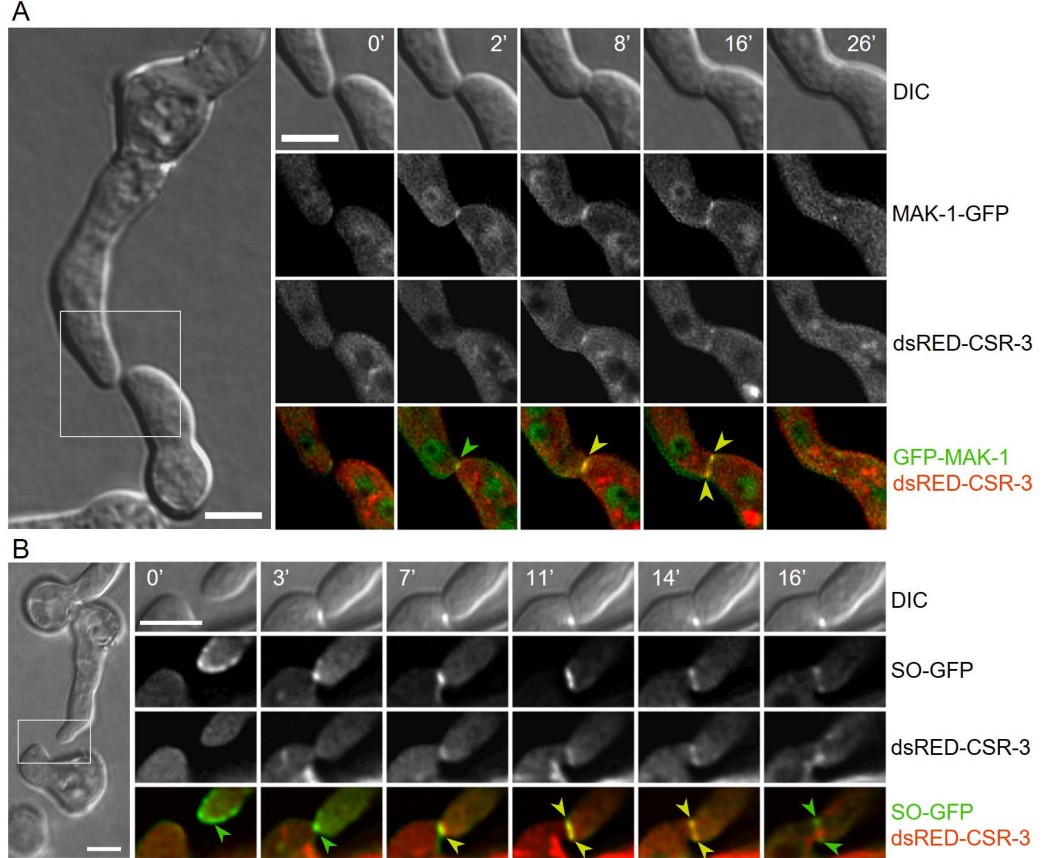

**Fig 8. CSR-3, the protein SO and the MAP kinase MAK-1 co-localize during fusion pore formation. (A)** Co-localization of dsRED-CSR-3 and GFP-MAK-1 during germling fusion within a heterokaryon isolate of the strains SH_95 (*Pccg-1-dsRed-csr-3*, Δ*csr-3*) and GN4-32 (*Pccg-1-gfp-mak- 1*, Δ*mak-1*). **(B)** Co-localization of dsRED-CSR-3 and SO-GFP in interacting germlings of a heterokaryon isolate of the strains SH_94 (*Pccg-1-dsRed-csr-3*, Δ*csr-3*) and N1-22 (P*ccg-1-so-gfp*). Scale bars: 5 µm.

## The chitin synthase CHS-2 displays a putative target of CSR-3

So far, the targets of CSR-3 remained unknown. An earlier study indicated that all of the seven described CHSs of *N. crassa* localize to developing septa, but they occupy different areas of the ingrowing membrane [32]. While CHS-1 (CHS class III) and CHS-3 to 7 (CHS class I, IV, V, VI, VII respectively) cover the complete septal region, CHS-2 (NCU05239) concentrates in a contracting ring like structure reminiscent to the localization pattern of CSR-3 during septation found in this study. We therefore hypothesized that CHS-2 might be a potential target of CSR-3. To further test this hypothesis, we co-expressed ds-Red-CSR-3 and CHS-2-GFP and found co-localization of both factors during septa formation in mature hyphae (Fig 10). Interestingly, the recruitment of CSR-3 to the contact site and during septation did not require the presence of CHS-2 (S13C, S13D Fig). When analyzing mature hyphae, we also noted that both proteins accumulate at septal plugs sealing the septal pore after hyphal injury. Here, the absence of CHS-2 caused slightly reduced recruitment of CSR-3 (Figs 10 and S13A, S13B).

When tested in fusing germlings, no CHS-2-GFP signal was detected at the fusion point. To test a potential contribution of CHS-2 to fusion pore formation, we therefore determined the lysis rate of fusion pairs in a Δ*chs-2* mutant. Comparable to the Δ*csr-3* mutant, the lysis rate was significantly increased on calcium poor medium. Combining both gene knockouts in a double mutant, whose macroscopic phenotype was indistinguishable to the Δ*csr-3* single mutant, did not result in

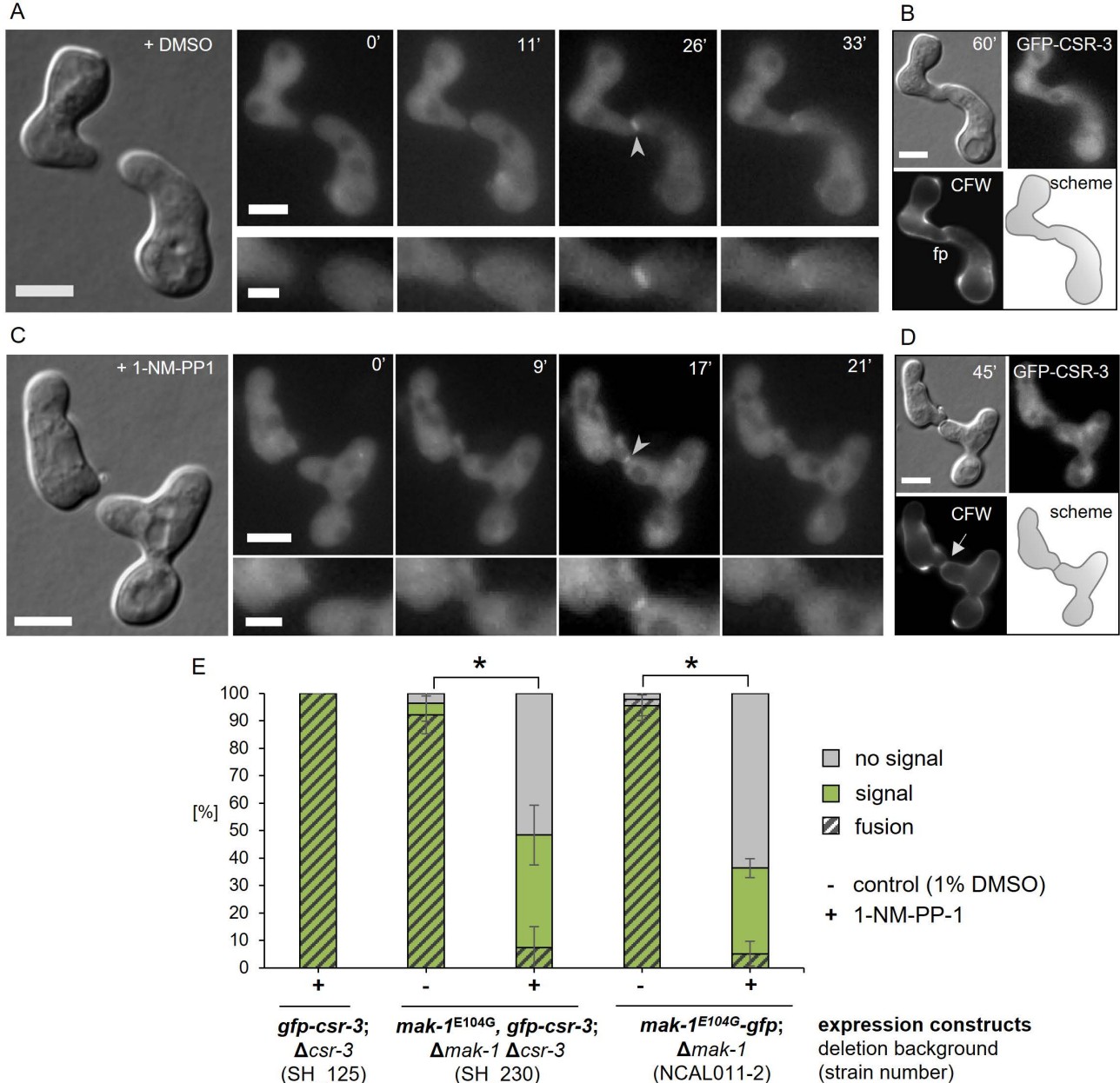

**Fig 9. The recruitment of CSR-3 to the contact site may depend on MAK-1.** (A,B) Localization of GFP-CSR-3 in interacting germlings (SH_230: *Pccg-1-gfp-csr-3*, *mak-1*[E104G], *Δcsr-3*, *Δmak-1*) under treatment of DMSO (A). Staining the cells 30-40 minutes after fusion with CFW revealed the successful formation of a fusion pore (fp)(B). (C,D) Localization of GFP-CSR-3 (SH_230: *Pccg-1-gfp-csr-3*, *mak-1*[E104G], *Δcsr-3*, *Δmak-1*) in interacting germlings after the inhibition with 1-NM-PP1 (B). Staining the cells 30-40 minutes after fusion with CFW revealed still cell wall material at the contact site (arrow) (D). scale bars: 5 µm (overview) and 2 µm (inset). (E) Quantification of localization of GFP-CSR-3 (SH_125: *Pccg-1-gfp-csr-3*, *Δcsr-3* and SH_230: *Pccg-1-gfp-csr-3*, *mak-1*[E104G], *Δcsr-3*, *Δmak-1*) and of MAK-1[E104G]-GFP (NCAL011-2: *mak-1*[E104G]-*gfp*, *Δmak-1*) at the contact site of interacting germling pairs (green bars). Germlings were treated with 1% DMSO and 1-NM-PP1. Only cell pairs without established contact before treatment were considered. After one hour, cells were stained with CFW to distinguish between fused and non-fused pairs. 4 to 20 individual pairs were examined in three independent experiments and statistically significant differences are indicated by asterisks.

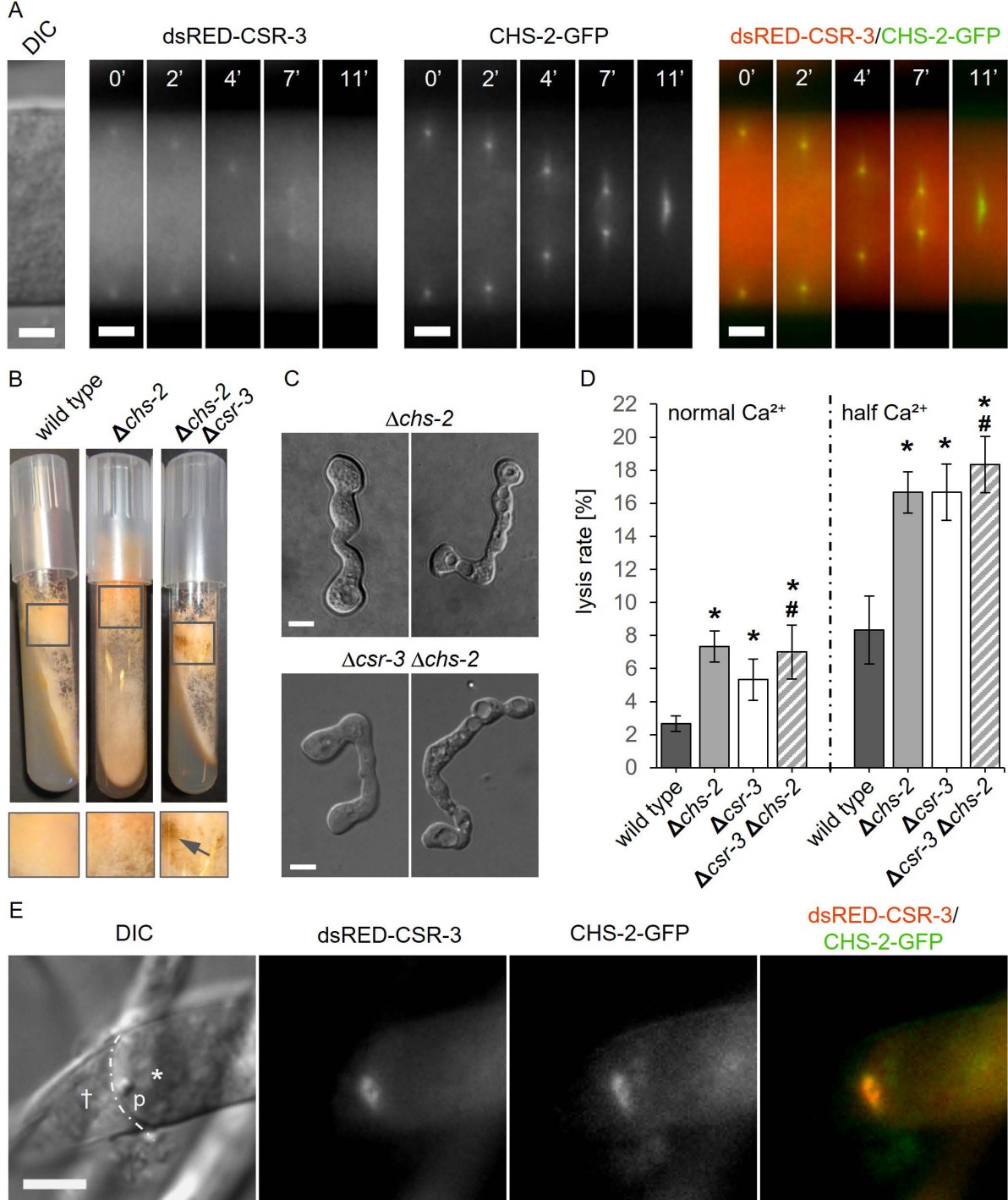

**Fig 10. CHS-2 and CSR-3 act in the same pathway. (A)** CHS-2-GFP and dsRED-CSR-3 co-localization during septation in a heterokaryotic isolate of the strains SH_95 (*Pccg-1-dsRed-csr-3*, Δ*csr-3*) and SH_157 (*Pccg1-chs-2-gfp*). Scale bar: 3 μm. **(B)** Macroscopic phenotypes of the wild type strain (FGSC 2489), the Δ*chs-2* mutant (FGSC 22804) and the double mutant SH_146 (Δ*chs-2*, Δ*csr-3*). Strains deficient of CSR-3 show a brownish pigmentation (arrows). **(C)** Healthy (left) and lysed (right) germling pairs of the Δ*chs-2* single mutant (FGSC 22804) and the double mutant SH_146 (Δ*chs-2*, Δ*csr-3*). **(D)** Quantification of the lysis rate during germling fusion of the wild type strain (FGSC 2489), the Δ*csr-3* (GN5-20) and Δ*chs-2* (FGSC 22804) single mutants, and the double mutant Δ*chs-2*, Δ*csr-3* (SH_146) on MM with half Ca²⁺. Error bars represent the standard deviation from three independent experiments (n = 100 each). The asterisks indicate statistically significant differences (p ≤ 0.05) compared to the wild type strain (FGSC 2489). No

significant additive effect of the double deletion compared to the single mutations was found (#). **(E)** DsRED-CSR-3 and CHS-2-GFP co-localization to the plug (p) separating the healthy part of a hypha (asterisk) from the injured one (cross) (heterokaryon isolate of the strains SH_95 (*Pccg-1-dsRed-csr-3*, Δ*csr-3*) and SH_157 (*Pccg1-chs-2-gfp*, Δ*csr-3*)). Scale bar: 10 μm.

a further increase of the lysis rate, suggesting that both proteins function in the same cellular pathway (Fig 10C, 10D). Taken together, our observations suggest that CHS-2 is a likely target of CSR-3.

## Discussion

The combined results of our gene knock mutant analysis and protein localization studies indicate that the *Neurospora crassa* chitin synthase regulator CSR-3 plays a role in various developmental processes involving cell wall synthesis and remodeling. These include septation, conidia formation, cell wall repair, and cell fusion. However, CSR-3 is not essential for most of these processes, supporting the view that cell wall synthesis and reconstruction involve a complex network of regulators and enzymes with both redundant and specific functions. For example, while none of the chitin synthases in *N. crassa* are essential, each contributes differently to cell wall regulation at distinct developmental stages and in various cell types [32]. A specific function of CSR-3 is its role in fusion pore formation during vegetative cell fusion. While general fungal growth relies on a delicate balance between cell wall deconstruction and synthesis to enable cell expansion without compromising wall integrity, cell fusion requires complete but locally restricted cell wall dissolution at the fusion site. This localized breakdown enables the opposing plasma membranes to come into close contact, facilitating their merger and the formation of the fusion pore. Dudin and colleagues elegantly demonstrated that during fission yeast mating, hydrolases are very precisely delivered to the fusion point via actin and myosin V [33]. This focused delivery locally shifts the balance between cell wall deconstruction and synthesis, which enables growth of the mating projection, toward deconstruction, thereby enabling pore formation. The authors hypothesize that in the surrounding area, cell wall synthesis might counteract hydrolase activity, stabilizing the forming fusion pore. Our data strongly support this hypothesis, showing that in *N. crassa*, the absence of the chitin synthase regulator CSR-3 and/or its target CHS-2 leads to more frequent aberrant cell fusion, resulting in lysis and cell death. This fusion focus hypothesis in *S. pombe* does not require new enzymatic activities for pore formation but depends on a fine-tuned geometry change and the resulting balance shift between the same hydrolases and synthases that mediate the growth of the mating projections [33]. Prior to our study, only one report described the localization of cell wall remodeling enzymes at cell fusion sites in filamentous fungi. In *N. crassa*, the two putative glycoside hydrolases, BGT-1 and BGT-2, localize to polar-growing hyphal tips, forming septa, and marking cell contact sites during germling and hyphal fusion [34]. While this observation suggests that growth and fusion share some of the molecular machinery for cell wall remodeling, our data indicate that fusion pore formation in *N. crassa* depends not only on shifts in enzyme activity but also on a distinct set of enzymes compared to those used in polarized tip growth. CSR-3 is not detected at the growing and interacting cell tips of the fusion partners, but only accumulates at the cell-cell contact area upon physical contact, coinciding with the cellular shift from directed growth to cell fusion. Our earlier studies indicated that this shift in cellular programming relies on the MAK-1 MAP kinase pathway, homologous to the cell wall integrity MAP kinase module of baker's yeast. Similar to CSR-3, MAK-1 accumulates at the cell contact zone upon physical contact of the fusion partners. Inhibition of MAK-1 activity results in the failure of growth arrest after cell contact and the subsequent cell fusion [30]. The upstream signals indicating cell contact in filamentous fungi remain unknown. In bakers and fission yeast, mating type-specific agglutinins or mating pheromones are thought to mediate contact recognition and fusion focus formation [35,36]. However, in conidial germling or hyphal fusion, the two cells are genetically identical, and yeast agglutinin genes are not conserved in the *N. crassa* genome. Similar to the upstream activators, the MAK-1 targets mediating the switch from growth to fusion had also remained unknown. Our data provide strong evidence that CSR-3 is a

direct or indirect target of the MAK-1 MAP kinase. Both proteins co-localize at the fusion point and share the same temporal dynamics in accumulation and release from that cellular location. Inhibition of the MAK-1 kinase activity prevented stable CSR-3 accumulation, indicating an indirect or direct post-transcriptional regulation of CSR-3 through MAK-1. While the role of the cell wall integrity pathway in the transcriptional regulation of chitin synthase genes is well established in yeast and filamentous fungi, its role in posttranscriptional regulation is still poorly understood [14,37,38]. A notable example is the involvement of the MAP kinase Slt2 in the accumulation of the chitin synthase Chs3 (CHS class IV) at the bud neck during cell division in baker's yeast. In absence of the kinase, accumulation of the chitin synthase to the bud neck is reduced, due to a compromised function of the regulator Chs4 [39]. Chs4, in turn, interacts with the septin-binding factor BNI-4, which is the direct target of Slt2 [40,41]. Our finding that mutating the three known phospho-sites of CSR-3 does not affect its localization, while inhibition of MAK-1 significantly reduces CSR-3 accumulation at the fusion site, supports a similar model in *N. crassa,* where CSR-3 is not a direct substrate of the MAP kinase, but is rather indirectly regulated.

A study in *S. cerevisiae* identified the septin-associated protein kinase Gin4 as a direct interaction partner of Chs4. The phosphorylation of Chs4 is partially dependent, either directly or indirectly, on this kinase, and its absence results in Chs4 mislocalization [42]. While these findings provide valuable insights into the role of kinases in regulating chitin synthase function, the complexity and species-specific nature of chitin synthesis regulation make it impractical to directly transfer mechanistic insights between species or processes. Identifying and comparing the role and function of phospho-regulation in various chitin synthase regulators will therefore be of great interest in the future.

Another key feature for the proper functioning of CSR-3 is the Caax motif at its C-terminus, with the specific sequence CVVM, suggesting potential prenylation by a farnesyl transferase. A similar farnesylation has been described for Skt5 in *S. cerevisiae* [43]. Our targeted mutagenesis approach revealed that while this motif is not required for the specific localization of CSR-3 during septation or germling fusion, it is essential for its proper function.

The role of Skt5 prenylation has been studied with somewhat inconclusive and partially contradictory findings. One study proposed that prenylation is crucial for proper localization and membrane association, consistent with a common role of prenylation in protein membrane binding [44]. In contrast, two studies concluded that its primary function is facilitating interaction and activation of its target chitin synthase, Chs3 [43,45] (DeMarine et al., 1997; Grabinska et al., 2007). Our findings support the view that CSR-3 prenylation is more important for interaction with its target chitin synthase than for membrane association.

Our study identified the chitin synthase CHS-2 as a likely target of CSR-3. The respective gene knockout mutants exhibit a similar lysis phenotype during germling fusion, and the absence of both genes shows no additive effect, suggesting that the two proteins function within the same pathway. In *Saccharomyces cerevisiae*, the chitin synthase Chs3 localizes to the bud neck even in the absence of its activator Chs4 but remains inactive and is rapidly endocytosed [44]. We propose a similar role for CSR-3 in the posttranslational regulation of CHS-2 in *N. crassa*.

The subcellular localization of CSR-3 revealed a strong and transient association of the regulator with sites of cell wall damage caused by enzymatic degradation. Its rapid accumulation at wound sites suggests posttranslational regulation, resulting in the translocation of CSR-3. While the transcriptional activation of chitin synthase genes, partly mediated by the cell wall integrity MAP kinase pathway, is well established [14], posttranslational mechanisms controlling the recruitment of chitin synthases to sites of cell wall damage remain poorly understood.

In *S. cerevisiae*, cellular stress induces the translocation of Chs3 from internal storage compartments (chitosomes) to the plasma membrane. This process requires the activity of the Rho1 GTPase and the Pkc1 protein kinase but is independent of the cell wall integrity MAP kinase module [46].

Based on our observations, we hypothesize that CSR-3 is a key component of the regulatory mechanism that recruits and activates chitin synthases at sites of cell wall damage. Unlike during cell-cell fusion, the MAP kinase MAK-1 does not co-localize with CSR-3 at wound sites, aligning with the dispensability of its homolog in stress-induced Chs3 translocation in yeast. Interestingly, the SO protein co-localizes with CSR-3 at wound sites, suggesting a MAP kinase-independent role

for this potential scaffolding protein. Consistent with this proposed role of CSR-3, the knockout mutant showed reduced resistance when interacting with the mycoparasitic species *T. atroviride*, which produces cell wall-degrading enzymes during its pathogenic development. Interestingly, no differences between the Δ*csr-3* mutant and the reference strain were observed on medium containing the cell wall dye calcofluor. These findings suggest that mutant phenotypes may become more pronounced in complex natural settings compared to conditions that test the effects of individual factors in isolation. This finding highlights the need for test settings that better mimic environmental growth conditions to fully understand the roles and functions of individual genes and proteins in fungal growth and development.

Remarkably, we observed rapid translocation of CSR-3 to the cell periphery following plasma membrane damage caused by the plant defense compound tomatine. Tomatine disrupts membrane integrity by binding to sterols and forming pores. As saponins, including tomatine, are not known to affect fungal cell walls, we hypothesize that CSR-3 translocation is a direct response to membrane disruption.

The disruption of membrane integrity typically leads to increased intracellular calcium concentrations, triggering membrane repair mechanisms. For instance, we recently identified the calcium-binding protein PEF-1 as a key mediator of membrane repair in response to tomatine in *N. crassa* and *B. cinerea* [47]. In *S. cerevisiae*, *C. albicans*, and *Aspergillus nidulans*, calcium activates the transcription of chitin synthase genes in a calcineurin-dependent manner [37,48,49]. However, the posttranslational regulation of chitin synthesis by calcium remains unexplored and represents a critical avenue for future research.

## Materials and methods

### *N. crassa* strains and growth conditions

Fungal strains used and generated in this study are listed in S1 Table. Strains were cultivated at 30°C on liquid or solid Vogel's minimal medium (MM) [50]. Required supplements for auxotrophic strains or for selection of resistance markers were added to the media prior to (2 mg/mL histidine) or after (200 µg/mL hygromycin) autoclavation. Medium with reduced $Ca^{2+}$ was prepared as described in Palma-Guerrero *et al.* 2014. For sexual propagation strains were grown on Westergaard's medium [51].

### Plasmid and strain construction

To generate the *csr-3* deletion mutant, a gene replacement strategy was employed as described earlier [52]. The knockout cassette was assembled by Yeast Recombinational Cloning (YRC) using fragments produced by PCR with the following primer combinations: 629/630 (5' flank), 631/632 (3' flank) and 82/83 (hygromycin resistance cassette) (S2 Table). The replacement cassette was transformed into strain FGSC 9719. Transformants were verified by PCR and Southern blot analysis. Homokaryotic strains GN5-20 and GN5-21 and a histidine auxotrophic isolate (SH_1) were isolated from a cross between a primary transformant and a heterokaryon of FGSC 6103 and FGSC 4564

### Transformation of *N. crassa*

*N. crassa* strains were transformed by electroporation of vegetative spores as described earlier [25,53].

### Construction of GFP-fused and mutated variants of CSR-3

To express GFP-CSR-3 and CSR-3-GFP fusion proteins under the control of either the P*ccg*-1 promoter or the native promoter P*csr-3*, the *csr-3* open reading frame was amplified by PCR using the primer pairs 936/937, 1074/1075 and 644/645. For the n-terminal tagged construct the fragment was cloned into the expression vector pMF334-GFP [54], modified by Elizabeth Hutchison, UC Berkeley, USA through the restriction sites *Bgl*II and *Xba*I or *Apa*I and *EcoR*I. The c-terminal tagged fragment was cloned into vector pMF272 (Freitag et al., 2004) via the *Pac*I and *Xba*I sites. The resulting

plasmids were transformed into strain SH_1, resulting in SH_59/SH_125/SH_126 (*Pccg-1-gfp-csr-3, Δcsr-3*), SH_283 (*Pcsr-3-gfp-csr-3, Δcsr-3*) and SH_257 (*Pccg-1-csr-3-gfp, Δcsr-3*).

To replace the critical cystein with a serine in the Caax motif of CSR-3 a point mutation was introduced at position 3257 of the ORF, exchanging the thymine with adenine. Two fragments of the coding sequence of *csr-3* were amplified by PCR with primer pairs 1523/1525 and 1525/1229, recombined using YRC and integrated into vector pMF334-GFP resulting in the plasmids no. 814 and no. 838. Transformation into SH_1 resulted in strains SH_290 and SH_316.

To block phosphorylation at position 166 (T), 683 (S) and 692 (S) of the CSR-3 protein, the amino acids were replaced by V, A and A respectively by YRC based on three mutated fragments of *csr-3* (primer pairs: 1224/1236, 1219/1221 and 1231/1229). To mimic a constant phosphorylation, the same amino acid residues were replaced with E, D and D (primer pairs: 1234/1239, 1219/1233 and 1236/1229). Both constructs were fused with either the P*ccg-1* or the P*csr-3* promotor and n-terminal GFP. The final plasmids were transformed into SH_1 resulting in strains SH_171, SH_176, SH_314, and SH_316.

Strains for analyzing of CSR-3 dynamics under inhibition of MAK-1 were created by crossing SH_126 (*Pccg-1-gfp-csr-3*, Δcsr-3, mat a*) with strain AS-111-848 (*hph-Pccg-1-mak-1$^{E104G}$, his-3$^-$, Δmak-1, mat A*).

## Quantitative phenotypical analysis of aerial hyphal growth, growth rate and sporulation

The quantification of aerial hyphal growth, growth rate and spore production was performed as described in [23] using an inoculum of $10^5$ conidia and four independent replicates for each experiment.

## Live-cell imaging and quantitative assays

To quantify fusion and lysis rates of germlings, microscopy samples were prepared as described previously [23]. In brief, spores from a one-week-old pre-culture were harvested by washing with water and filtering to remove mycelial remnants. $3 \times 10^6$ spores were evenly spread on a standard Petri dish and incubated at 30 °C for 3–5 hours before analysis by light and fluorescence microscopy.

To determine recruitment of GFP-CSR-3 at the contact sites during fusion, germlings were incubated for 2.5 h. To analyze quantity of successfully fused germling pairs carrying the *csr-3* deletion, germlings of the strains SH_12-14 (*Δcsr-3 his-3$^+$:: Pccg-1-mCherry, mat A*) and SH_9-11 (*Δcsr-3 his-3$^+$:: Pccg-1-gfp, mat A*) were mixed on plates and surveyed as described in [8]. Wild type strains expressing either cytosolic GFP (N3-06) or mCherry (N3-07) were used as reference strains. Fusion induced lysis was evaluated after 4 h of incubation by calculating the rate of germling pairs with a strong vacuolized appearance [9].

Microscopy of mature hyphae was conducted as described in [25]. Quantifications of distances between and diameter of the septa as well as numbers of side branches within the first 600 μm of a leading hypha were conducted using bright field microscopy and CFW staining and the software Fiji.

The analysis and quantification of the recruitment of GPF-CSR-3 in response to stress reagents were performed as described in [23] with the following substances: lysing enzyme (Glucanex, from *Trichoderma harzianum*; Sigma-Aldrich, stock solution 30 mg in 1 mL SMC buffer) and tomatine (CAS-Nr. [17406-45-0], stock solution 0.5 mg in 1 mL ddH$_2$O and 5% DMSO). As controls, the solvents alone were added to the samples. Fluorescent images of germlings were taken for a timespan of 5 min after the addition of 3 μl (lysing enzyme, SMC) or 5 μL (tomatine, ddH$_2$O, 5% DMSO) of the substances to an agar slice of 1 cm$^2$ and incubation time of 5 min.

Settings of the fluorescent microscopy for live-cell imaging were as described in [6].

## Staining of the plasma membrane and cell wall

Cell membranes were stained with FM4-64 (Invitrogen- Molecular Probes) as described in [4] with a final working solution of 25 μM of which 10 μL were added to agar squares of 1 cm$^2$. Cell walls were stained with calcofluor white (CFW; fluorescence brightener 28, Sigma-Aldrich; working solution: 7 μg/mL in ddH$_2$O and 0.1% DMSO).

## Chemical inhibition of MAPK activity

Mutation of the glutamic acid at position 104 to a glycine of MAK-1 allows the directed inhibition of the activity of the kinase by adding the bulky ATP analog 1-NM-PP1 (TRC A603003). To inhibit MAK-1 activity during vegetative fusion, addition of the inhibitor (1-NM-PP1, TRC A603003) to cultures and microscopic analyzes were conducted as previously described [5,30].

## Sensitivity assay

The survey of the sensitivity of *N. crassa* towards CFW and tomatine was performed as described in Schumann *et al.* 2019. Assays were analyzed after incubation for 72 h and 120 h at 30 °C in darkness.

## Interspecies confrontation assay

To co-cultivate *Trichoderma atroviride* and *N. crassa*, $2 \times 10^5$ and $2 \times 10^4$ spores, respectively, from 7 days old cultures were dropped on opposite sides of a Petri dish with either MM or MM with half calcium concentration. The plates were incubated for 7 days under day/night rhythm in room temperature. For quantification, the radius of the growing *T. atroviride* colony was measured from the center of the inoculation spot to the hyphal front and to the edge of the lysed area given as diameter. The co-cultivations were performed with at least three independent cultures of each species.

## Statistical analysis

To evaluate statistically significant differences (* $p < 0.05$), quantitative data were analyzed using the unpaired two-tailed student's t-test of Excel. Calculation of the standard deviation was performed using the STDEV function for samples from Excel.

## Supporting information

**S1 Literature. Literature for supplementary materials.**
(PDF)

**S1 Fig. The chitin synthase regulator 3 accumulates at the contact sites of fusing *N. crassa* germlings. (A)** Subcellular localization of GFP-CSA-1 in fusing germlings over time (strain SH_250: *Pccg-1-gfp-csa-1*). **(B)** Subcellular localization of GFP-CSA-2 in fusing germlings of strain SH_248 (*Pccg-1-gfp-csa-2*) over time. **(C)** Localization of GFP-CSR-3 (arrow head) in strain SH_45 (*Pccg-1-gfp-csr-3*) during germling fusion. Scale bars = 10 μm. Time scale = minutes.
(PDF)

**S2 Fig. CSR-3 exhibits multiple conserved motifs and validation of the created *csr-3* deletion mutant by southern blot. (A)** Scheme of the CSR-3 protein with selected motifs. Domains were predicted using CDSEARCH/cdd-analysis from NCBI (conserved domain database CDD; NCBI;) (Lu *et al.* 2020) (grey). The three selected phosphorylation sites (green) are from a previous phosphoproteome study done with iTRAQ (Jonkers *et al.* 2014) and the prediction for prenylation motifs (blue) were performed by using PrePS (Prenylation Prediction Suite) (Maurer-Stroh and Eisenhaber 2005). **(B/C)** Southern blot analysis of *csr-3* deletion in strain FGSC 9719: genomic DNA from eight primary transformants and the recipient strain FGSC 9719 was digested with *Eco*RI and subjected to southern blotting. Hybridization was performed using the radioactively labeled entire gene replacement cassette. Detection of signals at around 5 and 7 kb indicate a successful replacement of *csr-3* with the *hph* knock-out cassette. A signal at 15 kb indicates non-transformed nuclei. The 4100 bp signal in the recipient strain and transformants results from hybridization of the trpC promoter in the probe - driving expression of the hygromycin resistance gene - with the phosphinothricin resistance cassette, which contains the

same promoter and is integrated at the mus52 locus. The asterisk marks the primary transformant used for purification by crossing, resulting in homokaryotic *csr-3* deletion mutants GN5-20 and GN5-21.
(PDF)

**S3 Fig. Impact of CSR-3 on vegetative colony establishment. (A)** Mature colonies of wild type (N1-02) and the Δ*csr-3* mutant (GN5-21) on agar plates (left) and details of the hyphal front (FGSC 2489; GN5-20) (right). **(B, C)** Macroscopic phenotype of wild type (FGSC 2489) and Δ*csr-3* mutants (GN5-20, SH_283, SH_290, SH_125). Complemented strains (SH_283: *Pcsr-3-gfp-csr-3, Δcsr-3;* SH_125: *Pccg-1-gfp-csr-3, Δcsr-3*) show no phenotypic differences from the reference strain whereas mutants (GN5-20: Δ*csr-3*; SH_290: *Pcsr-3-gfp-csr-3*SAAX, Δ*csr-3*) lacking or with non-functional CSR-3 exhibit a brownish pigmentation (arrows in enlarged selection) and reduced linear growth linear growth. **(D, E)** *Csr-3* deletion has no effect on conidiation or aerial hyphae length. Statistically significant differences ($p \leq 0.05$) are indicated by asterisks. For details of quantification, see materials and methods.
(PDF)

**S4 Fig. CSR-3 is dispensable for germination, tropic interaction and fusion rate in germlings. (A)** Time-lapse imaging at 1–4 h temporal resolution of germling populations of the wild-type strain (FGSC 2489) and the *csr-3* deletion mutant (GN5-20) on MM medium at 30°C. Scale bars = 20 µm. **(B)** Quantification of germination and tropic interaction at four different time points during incubation of the wild type strain (FGSC 2489) and the *csr-3* deletion mutant (GN5-20) on MM medium at 30°C revealed no statistically significant differences. n = 100 spores each.
(PDF)

**S5 Fig. Recruitment of GFP-CSR-3 under control of the native promotor in germlings and hyphae. (A)** Subcellular localization of GFP-CSR-3 (arrow heads) during germling fusion in strain SH_285 (*Pcsr-3-1-gfp-csr-3, Δcsr-3*). **(B)** Localization GFP-CSR-3 during septum formation in a hypha of strain SH_285 (*Pcsr-3-1-gfp-csr-3, Δcsr-3*). Scale bar = 5 µm (A), 2 µm (B). Time scale = minutes.
(PDF)

**S6 Fig. CSR-3 is dispensable for the development of vegetative hyphae. (A)** Sample illustration of examined sections and septa of vegetative hyphae (here wild type strain FGSC 2489). **(B)** Strains were stained with CFW to visualize the cell wall to measure the distance between and the diameter of septa. **(C)** Quantification of time needed to form a complete septum in the wild type strain (FGSC 2489) and the strain GN5-20 (Δ*csr-3*). **(D)** Quantification of germ tube length and septa at three time points in wild type (FGSC 2489) and *csr-3* deletion mutant (GN5-20) germling populations. Error bars represent the standard deviation calculated from at least three independent experiments (B,C: n = 6–15 hyphae each; D: n = 55–140 germlings each). Statistically significant differences ($p \leq 0.05$) are indicated by asterisks.
(PDF)

**S7 Fig. CSR-3 is recruited during vegetative conidiation. (A/B)** Subcellular localization of GFP-CSR-3 (SH_125: *Pccg-1-gfp-csr-3, Δcsr-3*) (arrow heads) during the formation of macroconidia within an early (A) and a late (B) time point. **(C)** Illustration of the localization pattern of CSR-3 during the development of asexual macroconidia. **(D)** Localization of GFP-CSR-3 (arrow heads) during microconidiation. **(E)** CSR-3 is recruited to the conidiation site (0') and released (3') during microconidium formation **(F)** Illustration of the localization pattern of C SR-3 during the development of asexual microconidia. Strains were grown on plates with water agar. Scale bars = 10 µm (overview) and 2 µm (insets). Time scale = minutes.
(PDF)

**S8 Fig. CSR-3 localizes in at the cell periphery in response towards cell wall and membrane stress. (A,B)** Recruitment of GFP-CSR-3 (SH_125: *Pccg-1-gfp-csr-3, Δcsr-3*) (arrow) to the burst tip (c) of a germling 2 minutes after addition

of lysing enzyme (A) and SMC buffer as a control (B) (for corresponding movie see S2 Movie). **(C,D)** Localization of GFP-CSR-3 (SH_125: *Pccg-1-gfp-csr-3*, Δ*csr-3*) (arrow) at the tip of a germling 5 minutes after addition of tomatine (C) and DMSO as a control (D). **(E)** Quantification of CSR-3 localization patterns in response towards cell wall stress (lysing enzyme) and membrane stress (tomatine). Striped bars represent lysed cells (standard deviation values can be found in S4 Table). For details see material and method. Scale bars: 5 µm. Time scale: minutes.
(PDF)

**S9 Fig. CSR-3 is recruited dynamically to the tips of germlings treated with tomatine.** 48 minutes time-lapse of subcellular localization of GFP-CSR-3 (SH_125: *Pccg-1-gfp-csr-3*, Δ*csr-3*) (arrow heads) after treatment with tomatine. Spores were cultivated in liquid MM, which is why they underwent a slight rearrangement after addition of the substance. For experimental details see material and method. Scale bars: 5 µm (overview) and 2 µm (inset).
(PDF)

**S10 Fig. Sensitivity-profile of *N. crassa* wt and Δ*csr-3* mutant to different membrane or cell wall attacking substances. (A,B)** 10-fold serial dilution of conidial suspensions of wild type (FGSC 2489) and the *csr*-3 deletion mutant (GN5-20) were spotted on BDES medium containing the relevant drug respectively or a drug free control. The growth differences were checked after incubation for 72 h and 120 h at 30 °C in darkness (for details see material and methods).
(PDF)

**S11 Fig. CSR-3 promotes protection against the mycoparasite *T. atroviride*. (A)** Co-cultivation of *T. atroviride* (CBS 122147) wild type strain with either *N. crassa* wild type (FGSC2489), Δ*csr-3* mutant (GN5-20) or complemented mutant strain (SH_ 283: *Pcsr-3-gfp-csr-3*, Δ*csr-3*) on plates with MM after 4 days of incubation. **(B)** Quantification of the radius of the *T. atroviride* colony and the diameter of the lysed area within the *N. crassa* colony after 4 days of incubation. **(C)** Co-cultivation of *T. atroviride* (CBS 122147) wild type strain with either *N. crassa* wild type (FGSC2489), Δ*csr-3* mutant (GN5-20) or complemented mutant strain (SH_ 283: *Pcsr-3-gfp-csr-3*, Δ*csr-3*) on plates with MM with only half calcium concentration after 4 days of incubation. **(D)** Quantification of the radius of the *T. atroviride* colony and the diameter of the lysed area within the *N. crassa* colony after 4 days of co-incubation of three independent samples. Statistically significant differences (p ≤ 0.05) are indicated by asterisks. For details of quantification, see materials and methods.
(PDF)

**S12 Fig. Under treatment with 1-NM-PP-1 CSR-3's localization pattern is unaffected in a non-inhibitable background. (A)** Localization of GFP-CSR-3 (SH_125: *Pccg-1-gfp-csr-3*, Δ*csr-3*) during fusion pore formation (arrow heads) in interacting germlings under treatment with 1-NM-PP-1. **(B)** Staining the cells with CFW revealed the successful formation of a fusion pore (fp) and septa (s). **(C)** Localization of MAK-1$^{E104G}$-GFP during fusion pore formation (arrow heads) in interacting germlings (NCAL011-2: *mak-1$^{E104G}$-gfp*, Δ*mak-1*) under treatment with DMSO. **(D)** Staining the cells with CFW revealed the successful formation of a fusion pore (fp). **(E,F)** Addition of 1-NM-PP-1 prior to fusion results in the recruitment of MAK-1$^{E104G}$-GFP in some pairs (E), but nevertheless pairs don't succeed in fusion pore formation visible by remaining cell wall (arrow) and continues growth of germ tubes (F). For experimental details see materials and method.
(PDF)

**S13 Fig. CSR-3 localization pattern is not affected in a Δ*chs-2* mutant. (A,B)** Recruitment of GFP-CSR-3 (SH_297: *Pccg-1-gfp-csr-3*, Δ*chs-2*) (arrow head) to the plug (p) of injured hyphae (A) and its corresponding quantification in comparison to the strain SH_125 (*Pccg-1-gfp-csr-3*, Δ*csr-3*) (B). Scale bar: 10 µm. **(C)** 3D-Z projection of a stack of 45 single images of GFP-CSR-3 during septum formation of strain SH_297 (*Pccg-1-gfp-csr-3*, Δ*chs-2*). Scale bar: 3 µm. **(D)** Recruitment of GFP-CSR-3 (SH_297: *Pccg-1-gfp-csr-3*, Δ*chs-2*) to the inner ring of a forming septum within a hypha. Membrane was stained with FM4-64. Scale bar: 2 µm. **(E,F)** Localization of GFP-CSR-3 (arrow heads) during germling fusion in a

Δ*chs-2* mutant (SH_297: *Pccg-1-gfp-csr-3*, Δ*chs-2*) (E) and its corresponding quantification in comparison to the strain SH_125 (*Pccg-1-gfp-csr-3*, Δ*csr-3*) (F). Scale bar: 5 µm. Error bars represent the standard deviation from at least three independent experiments (B: n = 15–25 hyphae each; F: n = 15–20 germlings each). Statistically significant differences (p ≤ 0.05) are indicated by asterisks.
(PDF)

**S1 Table. Strains used in this study.**
(PDF)

**S2 Table. Oligonucleotides used in this study.**
(PDF)

**S3 Table. Plasmids used in this study.**
(PDF)

**S4 Table. Standard Deviations for S8 Fig.**
(PDF)

**S1 Movie. CSR-3 localizes in an expanding ring-like structure during vegetative fusion.** Subcellular localization of GFP-CSR-3 in strain SH_125 (*Pccg-1-gfp-csr-3*, Δ*csr-3*) during germling fusion over a time frame of 40 minutes. Combined movie of single pictures taken every 30 seconds of DIC (left) and fluorescence (right) filters. Time: minutes.
(AVI)

**S2 Movie. CSR-3 localization is highly dynamic during treatment with lysing enzyme.** Time lapse of the subcellular localization of GFP-CSR-3 (asterisks) in a liquid culture (SH_125: *Pccg-1-gfp-csr-3*, Δ*csr-3*) after treatment with lysing enzyme (added at minute 1). A forming septum (s) is observed at time point 30'. Combined movie of single pictures taken every 60 seconds of DIC (top) and fluorescence (bottom) filters. Time: minutes.
(AVI)

## Acknowledgments

We kindly thank Martin Weichert and Antonio Serrano for providing *N. crassa* strains for localizing MEK-1 and the inhibition assay of MAK-1. We are grateful to Laura Köpping for support in construction of the Δ*csr-3* mutant.

## Author contributions

**Conceptualization:** Stephanie Herzog, André Fleißner.

**Data curation:** Stephanie Herzog, Tanja N. Sedlacek, Kristian D. R. Roth, Manuel Reuning, André Fleißner.

**Formal analysis:** Stephanie Herzog, Tanja N. Sedlacek, Kristian D. R. Roth, Manuel Reuning, André Fleißner.

**Funding acquisition:** André Fleißner.

**Investigation:** Stephanie Herzog, Tanja N. Sedlacek, Kristian D. R. Roth, Manuel Reuning, Ulrike Brandt, Andre Fleißner.

**Methodology:** Stephanie Herzog, Tanja N. Sedlacek, Kristian D. R. Roth, Manuel Reuning, Ulrike Brandt.

**Project administration:** Stephanie Herzog, André Fleißner.

**Resources:** André Fleißner.

**Supervision:** Stephanie Herzog, André Fleißner.

**Validation:** Stephanie Herzog, Tanja N. Sedlacek, Kristian D. R. Roth, Manuel Reuning, André Fleißner.

**Visualization:** Stephanie Herzog.

**Writing – original draft:** Stephanie Herzog, André Fleißner.

**Writing – review & editing:** Stephanie Herzog, Tanja N. Sedlacek, Kristian D. R. Roth, Manuel Reuning, Ulrike Brandt, André Fleißner.

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
