## [Decision Letter · Decision Letter 0]

30 Jul 2025

PGENETICS-D-25-00651

The chitin synthase regulator CSR-3 promotes cellular integrity during cell-cell fusion in the filamentous ascomycete fungus Neurospora crassa

PLOS Genetics

Dear Andre,

Thank you for submitting your manuscript to PLOS Genetics. Three experts in the field have reviewed your manuscript. I agree with them in that your study is novel, rigorously conducted, and that you revealed important new information about fungal cell biology. You'll see below that the reviewers do not require additional experimentation; rather, all comments relate to updating the writing to improve understanding. We invite you to submit a revised version of the manuscript that addresses all points raised during the review process but I do not expect to send this back out to the reviewers.

Few reviewer comments stand out as requiring highlighting in this first part of the decision letter but there are three points I wish to stress. First, I agree with Reviewer 2 that some of the structures shown look like CATs; please comment on plating density of spores and whether you can ascertain that these are CATs or "standard" germ tubes. Second, I agree that when writing about chitin synthases one should be explicit because the naming conventions change from organism to organism. Third, though many folks claim that genes driven by the *ccg-1/grg-1 * promoter are overexpressed, unless you have data to that effect (by RT=PCR or northern) it is difficult to ascertain this. While the *grg-1/ccg-1 * gene is highly expressed in aerial hyphae and spores it tends to resemble a housekeeping gene in liquid and high glucose medium.

Lastly, while folks refer to *N. crassa*  as a "bread mold" this statement can be dropped without harm, especially as we don't even seem to be able to agree on the color ("red", "orange", and I know it as the "pink" bread mold). The name was originally given by folks who were looking at *N. sitophila* , so perhaps just call it Neurospora? 

-----------------

Please submit your revised manuscript within 30 days Aug 29 2025 11:59PM. If you will need more time than this to complete your revisions, please reply to this message or contact the journal office at plosgenetics@plos.org. Please include the following items when submitting your revised manuscript:

We look forward to receiving your revised manuscript.

Kind regards,

Michael Freitag

Academic Editor

PLOS Genetics

Geraldine Butler

Section Editor

PLOS Genetics

Aimée Dudley

Editor-in-Chief

PLOS Genetics

Anne Goriely

Editor-in-Chief

PLOS Genetics

**Journal Requirements:**

2) We noticed that you used the phrase 'data not shown' in the manuscript. We do not allow these references, as the PLOS data access policy requires that all data be either published with the manuscript or made available in a publicly accessible database. Please amend the supplementary material to include the referenced data or remove the references.

- ® on pages: 12, and 28

- TM on page: 28.

5) We have noticed that you have uploaded Supporting Information files, but you have not included a list of legends. Please add a full list of legends for your Supporting Information files after the references list.

Potential Copyright Issues:

i) Please confirm (a) that you are the photographer of 10B, S3, S6, and S11, or (b) provide written permission from the photographer to publish the photo(s) under our CC BY 4.0 license.

ii) Figures 2C, 2E, and 3F. Please confirm whether you drew the images / clip-art within the figure panels by hand. If you did not draw the images, please provide (a) a link to the source of the images or icons and their license / terms of use; or (b) written permission from the copyright holder to publish the images or icons under our CC BY 4.0 license. Alternatively, you may replace the images with open source alternatives. See these open source resources you may use to replace images / clip-art:

**Reviewers' comments:**

Reviewer's Responses to Questions

Reviewer #1: This is a very well written manuscript describing a thorough investigation on the role of CSR-3 in cell wall remodeling events. There are only very minor changes that this reviewer recommends. In referring to the prenylation signal the manuscript uses CAAX and CaaX designations. Since the a's refer to hydrophobic amino acids, I think the authors mention to use Caax throughout the manuscript. The authors will want to double-check spellings - I think the phosphoproteom on line 350 should be phosphoproteome. These are very minor issues and the reviewer comments the authors on an excellent study and manuscript.

Reviewer #2: The article “The chitin synthase regulator CSR-3 promotes cellular integrity during cell-cell fusion in the filamentous ascomycete fungus Neurospora crassa” by Herzog et al., is very well-written and presents a substantial amount of high-quality data aimed to discern the localization and role of the chitin synthase regulator CSR-3 during several developmental stages in N. crassa. The article provides valuable information in an area of cell wall fungal biology that is understudied. While all the experiments are conducted carefully and results are presented very elegantly, several points require clarification or revision to improve precision and accessibility for a broader scientific audience:

- The authors refer to Neurospora crassa as the red bread mold (Page 3, Line 50). It is more widely referred in the literature to as the orange bread mold.

- Page 3, line 58: The text should read CHS-2

- Page 4, lines 77-78: The wording is a bit confusing. What do the authors mean by “not only mating fusion”? And by “identical cells at two distinct developmental stages”?

- Page 4, line 81: What does the phrase “hyphal branches fuse to create cross connections” mean? Unclear what the cross connections refer to.

- Page 5, line 94: Referring to th cell wall as an organelle is inaccurate, as organelles are typically membrane-bound. It would be more appropriate to describe it as a structure.

- The nomenclature for chitin synthases is inconsistent across fungal taxa. It would be helpful if the authors specified the class of chitin synthases when referring to a specific CHS?

- Page 6, line 119: Why do the authors say chitin synthases contribute to cell wall remodeling? What about biosynthesis?

- S2A Fig: Unclear why all transformants exhibit a signal at 4100 bp, apparently because of the mus52 deletion in strain FGSC9719. Please provide a brief explanation.

- Page 8, line 168: Please add “in the csr-3 knockout strains” after “were unaffected”

- Page 9, lines 206-208: Under which conditions was the Pccg-1used to overexpress constructs?

- Fig. 2B: Is the structure shown a CAT (conidial anastomosis tube) rather than a germ tube? There are similar structures throughout the manuscript (Fig. 7 B and C for instance) that appear to be CATS. There is no mention of this in the article. Please clarify and discuss the significance.

- Fig. 2D, E: What is the nature of the expanding ring-like structure shown? This is commonly seen in N. crassa. Please elaborate.

- Several figures are missing scale bars, particularly in magnified insets (For example, Figs. 2, 3, 4, 5B, E, 6, 7, 8, 9, and S8, S9, S12 Figs. etc.). Please ensure all images include scale information.

- Fig. 6A: Please indicate what the sample was stained with.

- Page 11, line 246: Was the observed hyphal injury natural or experimentally induced? Please specify.

- Page 11, line 252: Please provide a reference for microconidia formation.

- Page 12, lines 274-275: In which figure is GFP-CSR-3 seen accumulating at the cell periphery?

- S8 Fig: The term should be “cytoplasmic”, not “cytoplasmatic”

- Page 13, line 294: “Therefor” should be “Therefore”

- Page 13, lines 301-304: Please rewrite for better clarity.

- Page 14, line 312: Please rewrite to improve readability.

- Page 14, line 316: S11 Fig, should be S11 A,B Fig; line 318 add (S11 C, D Fig) after calcium concentration.

- Page 15, line 350: Correct “phosphoproteom” to “phosphoproteome”

- Revise Fig. 7 legend. All panels A, B, and C mention constructs based on mik-1, whereas panel B appears to be based on mek-1, and panel C on mak-1. Please correct this for accuracy.

- Is it GFP-MAK-1 or MAK-1-GFP? Figure 7 and Figure 8 show different versions. Interestingly, in both figures MAK-1 tagged with GFP is also observed at what looks like nuclei. The authors do not mention this. Could it be relevant? Also, in figure 8, dsRed-CSR-3 appears at some globular organelles, not observed when imaging the GFP version. What are these organelles? Could the authors comment on this?

- MAK-1 and CSR-3, or SO and CSR-3 do not seem to co-localize during the entire duration of fusion pore formation based only on the visual evaluation of Figure 8. In some panels there seems to be signal of one protein but not the other, indicative of sequential appearance. Could the authors do a quantitative co-localization analysis using image analysis software to have more informative data? It would strengthen the observations.

- In Figure 9 legend: For Pccg-1-gfp-csr-3, mak- 1E104G, Δcsr-3, Δmak-1 and so on, could the authors organize the genotypes so that it is clearer for readers which deletion strain was complemented by which construct? What are the round organelles depicted in the schemes? Why some drawn but not others?

- Page 18, lines 425-426: This sentence is unclear. Do the authors mean that localization of CSR-3 did not require CHS-2? Please rephrase and elaborate on the reasoning.

- Page 19, first paragraph: What was the lysis rate on the mutant under normal calcium conditions?

- The supplementary movies play too quickly, making it difficult to follow events such as cytoplasmic leakage or fluorescence changes. Could the authors provide versions that run slower?

- Discussion:

- page 20

o Line 468: The fusion focus hypothesis? What do you mean?

o Lines 471-474. The authors state that prior to their studies no cell wall remodeling enzyme mediating cell fusion in filamentous fungi had been identified. Check Martinez-Núñez and Riquelme, 2015. Fungal Genetics and Biology. http://dx.doi.org/10.1016/j.fgb.2015.11.001. There the authors found BGT enzymes (GPI-anchored glycoside hydrolases) with a role in cell wall remodeling at fusion pores (Fig. 4B) and at septal pores (Fig. 2) with a similar distribution to CSR-3 and CHS-2.

- Line 657: Correct “Glukanex” to “Glucanex”

- It is hard to picture how cell wall biosynthesis is required for the formation of a stable fusion pore. A scheme illustrating the different steps leading to the pore formation and stabilization would be very helpful.

Reviewer #3: In this manuscript, Herzog et al present a detailed study of a chitin synthase regulator particularly involved in hyphal fusion. As the authors point out, hyphal fusion is a fundamental phenomenon in filamentous fungi, and is surprisingly understudied. Here, they show that CSR-3 particularly balances the cell-wall dissolution and repair necessary during fusion. Studies in this area are always difficult, due to the redundant nature of many of the molecular pathways, and the authors should be commended for undertaking such ambitious work. This work is important, as the formation of a hyphal network is nearly ubiquitous across fungi. Our limited understanding of how and why this network forms demands additional study, and this work provides essentials steps in this direction.

Regarding the manuscript itself, the claims are balanced, and accomodate the nuance of the results (I particularly like the use of the word "consistent with" when alternate explanations are possible). They appropriately situate their work in the field, and try to integrate their results with previous work of their own group and others. Finally, the manuscript is exceptionally well written and clearly structured. This is a lot of work, with a lot of microscopy results to explain, and the authors have a structure that walks the reader through it one step at a time.

Overall, I would suggest acceptance of the manuscript as-is, with two things I would suggest to consider:

Discussion-

Line 467: I guess it is a question whether one can say deletion in CSR-3 results in frequent cell fusion defects. Clearly, with reduced Ca2+ the defect becomes more apparent, but since the lysis rates are relatively moderate, I still question how severe the consequences are.

Methods-

Some more details would be useful for the Southern Blotting validation. For example, what sequence was used as a probe? I assume some sort of PCR product of csr-3?

Also, Some small typos I noticed:

Figures:

Figure S3C, extra x-axis labels present?

line 150: CDSAERCH

line 161: created using a gene replacement strategy

line 250: conidiophors

line 335: missing word, should this say "a CSR-3 version"?, also it probably helps others if you specify the amino acid position of the cysteine

line 350: phosphoproteome, also in caption for Figure S2A

line 424: I would hesitate to say "perfect", since the overlap is not literally 100%

line 434: likewise, I would hesitate to say "identical", maybe instead "indistinguishable"

line 464: This sentence is current tense, but when referring to Dudin et al (Line 457) that sentence is in past tense.

line 657: glucanex

**Have all data underlying the figures and results presented in the manuscript been provided?**

Reviewer #1: Yes

Reviewer #2: Yes

Reviewer #3: Yes

PLOS authors have the option to publish the peer review history of their article (what does this mean? ). If published, this will include your full peer review and any attached files.

**Do you want your identity to be public for this peer review?** For information about this choice, including consent withdrawal, please see our Privacy Policy .

Reviewer #1: No

Reviewer #2: No

Reviewer #3: No

**Figure resubmission:**
---

## [Editor Report · Decision Letter 1]

24 Sep 2025

Dear Dr Fleissner,

We are pleased to inform you that your manuscript entitled "The chitin synthase regulator CSR-3 promotes cellular integrity during cell-cell fusion in the filamentous ascomycete fungus Neurospora crassa" has been editorially accepted for publication in PLOS Genetics. Congratulations!

Thank you very much for the detailed responses and the attention you paid to the reviewer comments. We believe that you addressed all concerns more than adequately and therefore elected not to send this version out to review again.

Yours sincerely,

Michael Freitag

Academic Editor

PLOS Genetics

Geraldine Butler

Section Editor

PLOS Genetics

Aimée Dudley

Editor-in-Chief

PLOS Genetics

Anne Goriely

Editor-in-Chief

PLOS Genetics

Comments from the reviewers (if applicable): N/A

**Data Deposition**

http://datadryad.org/submit?journalID=pgenetics&manu=PGENETICS-D-25-00651R1

**Press Queries**

---

## [Editor Report · Acceptance letter]

PGENETICS-D-25-00651R1

The chitin synthase regulator CSR-3 promotes cellular integrity during cell-cell fusion in the filamentous ascomycete fungus Neurospora crassa

Dear Dr Fleissner,

We are pleased to inform you that your manuscript entitled "The chitin synthase regulator CSR-3 promotes cellular integrity during cell-cell fusion in the filamentous ascomycete fungus Neurospora crassa" has been formally accepted for publication in PLOS Genetics! Your manuscript is now with our production department and you will be notified of the publication date in due course.

With kind regards,

Zsofia Freund

PLOS Genetics

On behalf of:
